# Inter-comparison of surface meltwater routing models for the Greenland Ice Sheet and influence on subglacial effective pressures

Kang Yang[1,2], Aleah Sommers[3], Lauren C. Andrews[4], Laurence C. Smith[5,6,7], Xin Lu[1,2], Xavier Fettweis[8], Manchun Li[1,2]

[1]School of Geography and Ocean Science, Nanjing University, Nanjing, China
[2]Jiangsu Provincial Key Laboratory of Geographic Information Science and Technology, Nanjing, China
[3]Climate and Global Dynamics Laboratory, National Center for Atmospheric Research, Boulder, CO, USA
[4]Global Modeling and Assimilation Office, NASA Goddard Space Flight Center, Greenbelt, MD, USA
[5]Institute at Brown for Environment and Society, Providence, RI, USA
[6]Department of Earth, Environmental & Planetary Sciences, Brown University, Providence, RI, USA
[7]Department of Geography, University of California, Los Angeles, Los Angeles, CA, USA
[8]Department of Geography, University of Liège, Liège, Belgium

*Correspondence to*: Kang Yang (kangyang@nju.edu.cn)

**Abstract.** Each summer, large volumes of surface meltwater flow over the Greenland Ice Sheet surface and drain through
moulins to the ice sheet bed, where it impacts subglacial hydrology and ice flow dynamics. Runoff modulations, or routing
delays due to ice sheet surface conditions, propagate to englacial and subglacial hydrologic systems and require accurate
assessment to correctly estimate subglacial effective pressures and short-term lags between surface meltwater production and
ice motion. This study compares hourly supraglacial moulin discharge simulations from three surface meltwater routing
models, (1) synthetic unit hydrograph, (2) bare ice component of surface routing and lake filling, and (3) rescaled width
function, for four internally drained catchments located on the southwestern Greenland ice sheet surface. Using surface
runoff from the Modèle Atmosphérique Régionale regional climate model (RCM), simulated variables used for surface
meltwater routing are compared among the three routing models. For each catchment, simulated moulin hydrographs are
used as input to the SHAKTI subglacial hydrologic model to produce corresponding subglacial effective pressure variations
in the vicinity of a single moulin. Two routing models, surface routing and lake filling and rescaled width function, which
require the use of a digital elevation model (DEM), are assessed for the impact of DEM spatial resolution on simulated
moulin hydrographs. Surface routing and lake filling is sensitive to DEM spatial resolution, whereas rescaled width function
is not. Our results indicate that the three surface meltwater routing models perform differently in simulating moulin peak
discharge and time to peak, with rescaled width function simulating later, smaller peak moulin discharges than synthetic unit
hydrograph or surface routing and lake filling. We also demonstrate that the seasonal evolution of supraglacial stream/river
networks can be readily accommodated by rescaled width function but not synthetic unit hydrograph or surface routing and
lake filling models. Overall, all three models produce more realistic supraglacial discharges than simply using RCM runoff
outputs without an applied routing scheme; however, there are significant differences in supraglacial discharge generated by
the three models tested. This variability among surface meltwater routing models is reflected in a series of simple idealized

subglacial hydrology simulations, which yield different diurnal effective pressure amplitudes depending on the applied surface meltwater routing model; however, the temporal mean effective pressure is relatively consistent across routing models.

## 1 Introduction

Large volumes of meltwater are routed through supraglacial stream/river networks on the Greenland ice sheet (GrIS) each summer (Smith et al., 2015). In temperate areas of the ice sheet, most of this surface meltwater is injected to the bed via moulins (Catania et al., 2008; Lampkin and VanderBerg, 2014; Smith et al., 2015; Yang and Smith, 2016; Koziol and Arnold, 2018), where it can modulate ice flow (Bartholomew et al., 2011; Palmer et al., 2011; Banwell et al., 2013; Hewitt, 2013; Andrews et al., 2014; de Fleurian et al., 2016). However, the role of the supraglacial system in controlling subglacial hydrology remains poorly studied to date (Flowers, 2018). In particular, there are limited constraints on the efficiency of surface meltwater routing (Smith et al., 2017), resulting in large uncertainties in hydrodynamic coupling with ice motion.

In most studies that investigate surface-to-bed meltwater connections and the behaviour of the subglacial hydrologic system, surface meltwater routing is either simplified or simply ignored, i.e., moulin hydrographs are estimated directly from RCM output. In these instances, internally drained catchments (IDCs) drain surface meltwater through supraglacial stream/river networks to terminal moulins or lakes (Yang and Smith, 2016). Then, RCM instantaneous grid cell runoff is simply summed over each IDC to estimate its corresponding supraglacial moulin discharge or to fill supraglacial lakes (McGrath et al., 2011; Fitzpatrick et al., 2014). This summed value is used to drive models of subglacial hydrologic system evolution and/or ice flow dynamics. Such a simplification may be appropriate for small IDCs or for very long-term studies; however, surface meltwater routing has been found to substantially modify ice surface runoff (Banwell et al., 2012; Banwell et al., 2013; Arnold et al., 2014), by altering the magnitude and timing of peak moulin discharge (Smith et al., 2017; Yang et al., 2018). As such, surface meltwater routing has strong potential to affect subglacial hydrologic system evolution and hydrodynamics, especially for large IDCs and short (i.e., diurnal) time scales. The current lack of representation of the surface meltwater routing leads to an insufficient understanding of surface-to-bed meltwater connections and ice dynamics, particularly on diurnal timescales. Therefore, constraints on IDC discharge can provide critical boundary conditions for studies of the subglacial hydrologic system.

In terrestrial systems, including ice sheet and glacier surfaces, water routing may be characterized by transport velocity ($v$), flow length ($L$), and transport time ($t$). Meltwater transport velocity and flow length can be estimated from ice surface topography (Arnold et al., 1998), with the distribution of transport times representing the hydrologic response of a particular supraglacial catchment to surface melt (Yang et al., 2018). Alternately, empirical unit hydrograph (UH) methods can be used to estimate the hydrologic response based on catchment shape and area alone. UH is "a transfer function that is widely used for modelling catchment runoff response to rainfall events for some unit duration and unit depth of effective water input"

(Smith et al., 2017), and thus enable derivation of moulin hydrographs through convolution of input RCM surface runoff with remotely-sensed UH parameters.

In order to route supraglacial meltwater on ice surfaces, Arnold et al. (1998) and Banwell et al. (2012) developed the DEM-based Surface Routing and Lake Filling (SRLF) model. SRLF assumes all meltwater is transported downslope, with Manning's open-channel flow equation used to calculate spatially variable meltwater transport velocities for every bare-ice DEM pixel.

More recently, the Snyder Synthetic Unit Hydrograph (SUH) (Snyder, 1938) was adapted for use on ice sheets to estimate moulin hydrographs without the need for a DEM (Smith et al., 2017). To do this, Smith et al. (2017) calibrated two key SUH parameters using a field-measured moulin hydrograph, and a Gamma function was then used to build individual SUHs for hundreds of remotely-sensed internally drained catchments (IDCs) (Smith et al., 2017). Because the Snyder SUH is a simple lumped model that does not rely on DEMs, it provides a straightforward but static approach to route surface meltwater to moulins.

Most recently, Yang et al. (2018) utilized Rescaled Width Function (RWF) (D'Odorico and Rigon, 2003) to further partition the bare ice surface into either interfluve (i.e., hillslope) or open-channel zones using high-resolution satellite imagery to discern between the two. Catchment-averaged meltwater transport velocities for each zone were then calibrated using a field-measured moulin hydrograph (Smith et al., 2017), with slow interfluve flow ($\sim 10^{-3}$-$10^{-4}$ m/s) and fast open-channel flow ($\sim 10^{-1}$ m/s) combined to simulate the downstream moulin hydrograph.

These three surface meltwater routing approaches (SRLF, SUH, and RWF) use different assumptions and have different data requirements (Table 1). SUH is the simplest model; it requires catchment shape, area, and several empirically-derived parameters to estimate surface meltwater transport time (Smith et al., 2017). In contrast, RWF requires a partitioning of slow (interfluve) and fast (open channel) flow that can substantially impact the performance of RWF without careful spatial and temporal calibration. Moreover, SRLF and RWF rely on surface DEMs to calculate meltwater flow paths (Yang et al., 2018) and SRLF also relies on DEMs to calculate meltwater routing velocities (Arnold et al., 1998). Characteristics such as slope, flow direction, flow length, drainage area, and drainage networks are readily extracted from DEMs but are scale-dependent (Montgomery and Foufoula-Georgiou, 1993), signifying that DEM spatial resolutions influence their computation (Zhang and Montgomery, 1994; Hancock et al., 2006). This DEM resolution dependence can now be tested with the advance of high-resolution DEM datasets (e.g., 2 m ArcticDEM) of the GrIS surface (Noh and Howat, 2015, 2017, 2018).

To assess supraglacial discharge variability among current surface meltwater routing models, this study simulates surface meltwater runoff through four moderate-size IDCs using the SUH, SRLF, and RWF models described above (Figure 1). Routing model ability to simulate meltwater transport velocity, flow length, transport time, UH, peak moulin discharge, and time to peak are compared, and the impact of DEM spatial resolution on SRLF and RWF is examined. The differences in using these models to drive subglacial effective pressure variations are qualitatively investigated with an idealized case of meltwater input to the bed via a single moulin, using the Subglacial Hydrology and Kinetic, Transient Interactions (SHAKTI)

subglacial hydrologic model (Sommers et al., 2018) to explore the potential impact of surface meltwater routing on subglacial effective pressure fluctuations.

Notably, this study cannot, in good faith, focus on method comparison, nor determine the "best" (i.e., best able to reproduce a real-world moulin hydrograph) model due to the lack of calibration and validation data. Owing to this limitation, the goal of this study is to assess differences among the three surface meltwater routing models, rather than to determine which model most realistically simulates surface meltwater routing on the ice surface. We conclude with a general discussion of surface meltwater routing on the Greenland ice surface and include some recommendations for best practices and future research.

## 2 Methods and Data

### 2.1 Study area and data sources

We selected four moderate-size supraglacial IDCs in the Russell Glacier region, southwestern GrIS to explore the impact of SUH, SRLF, and RWF meltwater routing models on moulin discharge and subglacial effective pressure (Figure 1, Table 2). They are distributed at approximately 200 m elevation intervals in order to span the elevational range of most well-developed IDCs found in the Russell Glacier region and the variable surface melt conditions of this region (Fitzpatrick et al., 2014; Yang and Smith, 2016). Large supraglacial lakes can store considerable volumes of surface meltwater and thus impact moulin discharge at the IDC outlet (Yang et al., 2019). Therefore, these four IDCs were deliberately chosen to avoid the presence of large supraglacial lakes, and all surface meltwater is routed to the terminal moulin at the catchment outlet (Figure 1). As such, surface runoff produced in each IDC should equal to the moulin discharge (Smith et al., 2017). The four IDC areas range from 53.0 $km^2$ to 66.9 $km^2$; surface elevations vary from 1086 m to 1672 m and ice thicknesses from 854 m to 1432 m (Table 2). A moulin discharge hydrograph collected at Rio Behar catchment (IDC2 in this study; 67.049346N, 49.025809W) for 72 h from 20 to 23 July 2015 was used to calibrate parameters of SUH and RWF models (see Sections 2.5 and 2.6). It is problematic to apply these empirically-derived parameters over large spaces and long times (Yang et al., 2018). Therefore, the selected IDCs are distributed in a relatively small region and the areas of IDC1, IDC3, and IDC4 are similar to the Rio Behar catchment (IDC2).

The high-resolution (2 m) ArcticDEM is the primary data source for our study of surface meltwater routing in the four supraglacial IDCs. ArcticDEM products are created from high-resolution (0.5 m) WorldView-1/2/3 stereo images by Polar Geospatial Center (PGC) and provide an unprecedented opportunity to investigate ice surface landscape and hydrologic processes (Noh and Howat, 2015, 2017, 2018). Supraglacial IDC boundaries were derived from ArcticDEM following the method of Karlstrom and Yang (2016). MAR (Modèle Atmosphérique Régionale) 3.6, a regional coupled atmosphere-land climate model with 20 km native horizontal resolution (Fettweis et al., 2013), provides hourly surface runoff simulations. Catchment-averaged hourly runoff [mm/h] was obtained by clipping MAR grid pixels with the supraglacial IDC boundaries

and summing their corresponding runoff values (Smith et al., 2017). Absent any routing, the resulting area-integrated runoff is termed RCM instantaneous runoff.

## 2.2 Unit Hydrograph

A surface meltwater routing model does not need to explicitly determine how meltwater produced on a cell is routed to its downstream grid cell(s) in a catchment, which is the aim of spatially-distributed routing models (Singh et al., 2014). In contrast, a routing model can be lumped and only determines how surface meltwater produced in the catchment is temporally routed to the catchment outlet. Unit hydrograph (UH) is designed for this purpose and provides a transfer function between surface runoff $R$ and moulin discharge $Q$ (i.e., direct hydrograph):

$$Q = R * \text{UH} \quad (1)$$

where * is the convolution operator. For each IDC, convolution of UH with hourly $R$ yields a discharge hydrograph at the catchment outlet (i.e., moulin) and thereby determines how surface meltwater produced in the catchment is temporally routed to the catchment outlet. Therefore, the three surface meltwater routing methods (SUH, SRLF, and RWF) based on UH can be considered routing models.

## 2.3 Snyder Synthetic Unit Hydrograph

Snyder SUH (Snyder, 1938) assumes that catchment morphometry controls the shape of UH and utilizes the catchment main-stem stream length ($L$, in km), the distance from the catchment outlet (here, the terminal moulin) to the point on the main channel nearest to the catchment centroid ($L_{ca}$, in km), and two parameters, $C_p$ and $C_t$, to determine peak discharge ($h_p$, in hr$^{-1}$) and time-to-peak ($t_p$, in hours) of SUH as:

$$t_p = C_t(LL_{ca})^{0.3} \quad (2)$$

$$h_p = C_p/t_p \quad (3)$$

$C_p$ and $C_t$ are two parameters depending on "units and drainage-basin characteristics" (Snyder, 1938). Smith et al. (2017) used a moulin discharge hydrograph collected at the Rio Behar catchment during the 20 to the 23 July 2015 to calibrate $C_p$ and $C_t$. Following Smith et al. (2017), we use $C_p = 0.72$ and $C_t = 1.61$ to determine $h_p$ and $t_p$ of SUHs and employ a Gamma distribution to determine the catchment-specific SUH for each IDC.

Although Smith et al. (2017) demonstrated parameter transferability using two other independently field-measured moulin hydrographs, the two calibrated parameters, $C_p$ and $C_t$, are collected at one moulin for one moment in time so parameter transferability over large regions and long timescales may still be limited. However, it is reasonable to apply them for the Russell Glacier region, southwestern GrIS during July 2015.

## 2.4 Surface Routing and Lake Filling

SRLF has been used to model filling of supraglacial lakes within supraglacial catchments (Banwell et al., 2012). Here we use the model to route surface meltwater downslope to a moulin and subsequently to the ice sheet bed. The SRLF model

employs Darcy's law to route surface meltwater flow through snow and Manning's open-channel flow equation to route meltwater flow over bare-ice surfaces (Arnold et al., 1998). SRLF is the first surface meltwater routing model for the GrIS and has been successfully applied to simulate supraglacial lake growth and to drive subglacial hydrological evolution during several complete melt seasons (Banwell et al., 2012; Banwell et al., 2013; Arnold et al., 2014; Banwell et al., 2016). In this study, we use its bare-ice component to make it comparable with the SUH and RWF (Smith et al., 2017; Yang et al., 2018) and representative of mid melt season conditions in the ablation zone. SRLF assumes universal presence of supraglacial open-channel flow everywhere on bare ice surface, by applying Manning's open-channel flow equation (Manning, 1891) to calculate channelized meltwater velocity $v$ for every pixel along a DEM-derived flow path:

$$v = R_H^{2/3} S^{1/2}/n \quad (4)$$

where $R_H$ is the hydraulic radius of the supraglacial meltwater channel, $S$ is water surface slope calculated from DEM, and $n$ is the Manning roughness parameter (Arnold et al., 1998). Following Arnold et al. (1998), we set $R_H = 0.035$ m and $n = 0.05$. We calculated $S$ per pixel using ArcticDEM. The shortest meltwater flow length ($L$) was determined from ArcticDEM using the D8 flow routing algorithm following Karlstrom and Yang (2016). Meltwater transport time ($t$) for each DEM pixel was then determined as $t = L/v$. To compare with SUH and RWF, a one-hour unit hydrograph was calculated by hourly binning the transport time raster (Yang et al., 2018), with the resultant UH termed SRLF UH.

## 2.5 Rescaled Width Function

The RWF routing model partitions surface meltwater transport pathways into interfluve (i.e., hillslope) and open-channel distances ($L_h$ and $L_c$), to determine two catchment-averaged meltwater velocities ($v_h$ and $v_c$) to represent routing efficiency over these regions (D'Odorico and Rigon, 2003). Then, the transport time for each pixel in a catchment may be calculated as:

$$t = t_h + t_c = L_h/v_h + L_c/v_c \quad (5)$$

where $t_h$ is the interfluve travel time and $t_c$ is the channel travel time. Similar to SRLF, the shortest meltwater flow path for interfluve and channel transport distances is determined using a DEM (Karlstrom and Yang, 2016), with the relative coverage determined by the extent of supraglacial streams and rivers (Yang et al., 2018). By applying the constant interfluve velocity $v_h$ to interfluve pixels and channel velocity $v_c$ to open channel pixels, the meltwater transport time from every catchment pixel to the moulin can be determined and a consequent catchment UH generated, hereafter referred to as RWF UH. The primary difference between RWF and SRLF is the way they determine meltwater flow velocity for each catchment cell. For RWF, using the in situ moulin hydrograph of Smith et al. (2017), Yang et al. (2018) determined the velocity combination of $v_h = 0.0006$ m/s and $v_c = 0.4$ m/s to be the optimal match to field observations. Although $v_h$ and $v_c$ may change notably during the melt season because of variable surface melt intensity and ice surface topography, it is reasonable to assume that $v_h$ and $v_c$ are persistent during a relatively short time period for similar hydrological and glaciological environments. Therefore, in this study, the optimal velocity combination obtained in (Yang et al., 2018) is assumed to be transferable and used to create RWF UHs for the four IDCs.

## 2.6 Controls of DEM resolution on surface meltwater routing

DEM spatial resolution has important impacts on the hydrologic response of terrestrial catchments to precipitation (Zhang and Montgomery, 1994; Hancock et al., 2006). To estimate the control of DEM spatial resolution on ice surface meltwater routing, 2 m resolution ArcticDEM data were resampled to 5 m, 10 m, 30 m, and 90 m resolutions, similar to Zhang and Montgomery (1994), and the corresponding meltwater flow paths, velocities, transport time, UHs, and moulin hydrographs for SRLF and RWF subsequently calculated. For SRLF, all five resampled DEMs were used. For RWF, only 2 m, 5 m, and 10 m resolutions were used to prevent DEM resolution from exceeding the typical interfluve transport distance, ~10 m for southwestern GrIS (Yang et al., 2018); otherwise, interfluve transport distance would be overestimated and the resultant hydrograph would be inappropriate (Hancock et al., 2006).

## 2.7 Temporal evolution of surface meltwater routing

The spatial pattern of supraglacial stream/river networks is known to vary significantly during a melt season (Lampkin and VanderBerg, 2014). This affects the hydrologic response (and UH) of a supraglacial IDC as open-channel and interfluve transport distances change (Montgomery and Foufoula-Georgiou, 1993; van Meerveld et al., 2019). RWF can mimic this effect by varying the partitioning of interfluve versus open-channel zones (Yang et al., 2018). The bare ice component of SRLF, in contrast, assumes the bare ice surface has a stable response to the surface melt and uses static meltwater routing velocities to build the UH. SUH relies on $C_p$ and $C_t$ to build the UH. If a moulin hydrograph is available during the entire melt season, we can calculate time evolving $C_p$ and $C_t$ using variable time-to-peak ($t_p$), peak discharge ($h_p$), and main-stem channel length ($L$), thus creating multi-temporal UHs. Without such direct measurements, the parameters cannot be realistically varied in time and SUH cannot mimic variable hydrologic response of a catchment to surface melt.

Capturing the rapid seasonal evolution of supraglacial stream/river networks is challenging (Lampkin and VanderBerg, 2014; Yang et al., 2017) and remains largely unperformed (Flowers, 2018), but a positive relationship exists between surface melt and supraglacial drainage density (Yang et al., 2017). As a simplified simulation, this study uses a series of cumulative contributing area (or channel initial threshold, $A_c$) values ($A_c$ = 100 m$^2$, 250 m$^2$, 500 m$^2$, 1000 m$^2$, 2500 m$^2$, 5000 m$^2$) to simulate the hypothetical seasonal evolution of supraglacial stream/river networks by evolving the partitioning of interfluve vs. open channels and resultant surface meltwater routing simulations (Yang et al., 2018). $A_c$ defines the surface area needed to initiate open channel flow. Smaller values of $A_c$ signify periods of intense surface melt and actively flowing, well-developed supraglacial stream/river networks (Smith et al., 2017), whereas larger values signify poorly developed network characteristics of the late melt season (Yang et al., 2018). A total of six 5-day RWF UHs were created for each $A_c$ value, broadly consistent with a decreasing RCM runoff trend during July 2015. Each RWF UH was used to calculate moulin discharge for five days and the resultant moulin discharge is termed as dynamic $A_c$ discharge.

**2.8 Subglacial hydrology modelling**

A variety of numerical models have been formulated to simulate different aspects of the subglacial drainage system and compute subglacial effective pressure (defined as the difference between ice overburden pressure and subglacial water pressure) (Flowers, 2015; de Fleurian et al., 2018). To examine the variations in modelled effective pressure resulting from the different surface meltwater routing models considered in this study, we treat their respective moulin discharges as direct meltwater inputs to the bed via a single moulin, and simulate subsequent evolution of the local subglacial drainage system in an idealized setting using the SHAKTI model (Sommers et al., 2018).

SHAKTI is built within the Ice Sheet System Model (Larour et al., 2012). Using finite elements, it applies a single set of equations over the entire model domain, conserving mass, energy, and momentum, with a subglacial water flux formulation allowing for local development of both turbulent and laminar flow regimes, as well as regimes corresponding to the broad transition between laminar and turbulent. SHAKTI calculates transient effective pressure, hydraulic head, subglacial gap height, subglacial water flux, and "degree of channelization" (defined as the ratio of the rate of opening by melt to the rate of opening by sliding over bumps in the bed). Subglacial effective pressure and geometry evolve naturally to produce continuous configurations ranging from sheet-like to channelized drainage. A thorough description of the SHAKTI model equations, methods, and features, including limitations, can be found in Sommers et al. (2018).

To explore and compare the influence of different surface meltwater routing models on idealized subglacial hydrology in a small experiment, we apply meltwater input at a single moulin at the center of a 1 km square domain. The domain is discretized with an unstructured triangular mesh consisting of 614 elements and a typical element edge length of 50 m. Outflow is to the left edge of the square domain, with a Dirichlet boundary condition setting the subglacial water pressure to 50 % of the ice overburden pressure (a reasonable assumption for our drainage catchments far from the ice sheet margin), and zero-flux pressure Neumann boundary conditions on the other three sides (i.e. enforcing *dh/dx=0* at the upstream right boundary and *dh/dy=0* at the top and bottom boundaries, where *h* is hydraulic head, which is directly related to water pressure). For each IDC, the mean surface slope and mean ice thickness are used (Morlighem et al., 2017) for the domain geometry, with the bed slope set equal to the surface slope to create a uniformly thick tilted slab of ice, and constant sliding velocity is prescribed as 100% of the annual mean observed surface velocity (Table 3). Note that two-way coupling is not currently enabled in SHAKTI/ISSM; the prescribed sliding velocity influences the subglacial system but the feedback of evolving effective pressure on sliding velocity is not implemented. Constants and parameter values used in the simulations are summarized in Table 3. As with all subglacial hydrology modelling efforts, some of these parameters are highly uncertain and poorly constrained (particularly with respect to parameters such as bed bump height and bed bump spacing). The values used in these experiments are based on values used in earlier work with SHAKTI (Sommers et al., 2018) and in the Subglacial Hydrology Model Inter-comparison Project (de Fleurian et al., 2018). There is a great need to better constrain uncertain parameters used in subglacial hydrology models, but that endeavour is beyond the scope of this work.

## 3 Results

### 3.1 Simulations of supraglacial moulin discharge

Inclusion of surface meltwater routing clearly modifies the timing of peak moulin discharge relative to RCM runoff obtained from MAR, with each of the three routing models performing somewhat differently. When RCM instantaneous runoff alone is used to simulate moulin discharge (i.e., no routing), moulin discharges peak between 13:00-15:00 local time for all the four studied IDCs (Table 2). Inclusion of routing processes, however, introduces considerable delay between the timing of peak RCM runoff generation and its arrival at the moulins, with all routing models displaying peak moulin discharges occurring from 17:00 to 23:00 local time (Table 2, Figures 2 and S1-S3).

SUH-routed hydrographs simulate peak moulin discharge between approximately 19:00 and 21:00, a delay of 6 hours compared to the RCM instantaneous peak discharge, while SRLF-routed delays vary from 3 hours to 10 hours and RWF-routed delay varies from 7 hours to 10 hours (Table 2). These delays are longer than ~2 hours measured for small IDCs near ice margins (Shepherd et al., 2009; McGrath et al., 2011; Bartholomew et al., 2012) but are comparable to ~5-7 hours measured for the ~60 km$^2$ Rio Behar catchment, consistent with findings indicating that catchment size controls surface meltwater routing delays (Smith et al., 2017).

Surface meltwater routing also controls peak moulin discharge magnitude and diurnal discharge range. This observation is clearly demonstrated when examining the derived UHs for each routing model. The UH isolates the impact of basin characteristics on moulin discharge by providing the basin integrated runoff resulting from one centimetre of surface melt over one hour. The resultant patterns of moulin hydrographs for the four IDCs are similar, so we use IDC1 to illustrate our results here. The three UHs (i.e. SUH, SRLF UH, and RWF UH) and their routed moulin discharges for the the other three IDCs are presented as supplementary material (Figures S1-S3). For IDC1, the peak values of RWF UHs are lower than SUHs and 2 m SRLF UHs. Therefore, RWF UHs temporally distribute surface meltwater most smoothly (Figures 2a, 2c, and 2e). The peak RWF UH is ~0.10 (Figure 2e), indicating that ~10 % surface meltwater is routed to the moulin during the peak runoff hour (Singh et al., 2014), while the peak values are ~0.13 for SUH (Figure 2a) and ~0.15 for 2 m SRLF UH (Figure 2c).

UHs with lower amplitude yield smaller peak moulin discharge magnitudes and diurnal discharge ranges. Therefore, inclusion of surface meltwater routing introduces smaller peak moulin discharge relative to the RCM instantaneous peak discharge (Figures 2b, 2d, and 2f). For IDC1, the peak moulin discharges simulated by the three routing models are >25 % lower than the RCM instantaneous peak moulin discharge, while the diurnal discharge ranges are >27 % lower. Among the three routing models, RWF introduces slightly smoother moulin hydrographs than SUH and 2 m SRLF (Figures 2b, 2d, and 2f), yielding small differences in peak moulin discharge (<7%) and diurnal discharge range (<12%). This finding suggests that all three models are more representative of the observed processes than using RCM instantaneous runoff without routing, but that each model collects and distributes surface meltwater with different efficiency.

### 3.2 Simulations of subglacial effective pressure

When applied to provide meltwater input to the bed via a single moulin, the SRLF, SUH, and RWF meltwater routing models all introduce differences in diurnal and long-term fluctuations in subglacial effective pressure (ice overburden pressure – water pressure) relative to effective pressure produced using RCM instantaneous runoff. The effective pressures presented are spatial mean effective pressures over the entire 1-km square domain of the bed surrounding the moulin input. The resultant temporal pattern of effective pressure simulation is similar among the four IDCs although the magnitude and amplitude differ substantially due to differences in ice thickness, slope, and sliding velocity of each catchment. We focus on IDC1 to illustrate our results here (Figure 3), with results for the other three IDCs presented in the supplement (Figures S4-S6). For IDC1, the temporal mean of the spatial mean effective pressure is relatively consistent between routing models, varying by < 4% (ranging from 3.36 MPa to 3.48 MPa). RCM instantaneous runoff produces a slightly higher temporal mean effective pressure of 3.53 MPa. The amplitude of the fluctuations around the mean, however, differs substantially between different routing models.

The relative magnitudes of diurnal signals seen in the meltwater inputs are reflected in the subglacial response. The effective pressure amplitude scales relative to the meltwater input amplitude, with the largest amplitude produced by the RCM instantaneous runoff inputs (Figure 3). While all surface meltwater routing models dampen these amplitudes as compared to RCM instantaneous runoff, the models that produce relatively high-amplitude moulin inputs correspond to high-amplitude effective pressure variations. In all cases a clear preferential pathway (i.e. a low "channel") develops from the moulin location to the outflow at the left edge of the domain, as expected, based on the topography and boundary conditions of the experiment design (Figure 4). This efficient drainage pathway evolves with the inputs through time and is characterized by higher gap height and effective pressure (lower hydraulic head and water pressure) than the surrounding bed perpendicular to flow (Figure 4, and supplementary animation Figure S7).

For all surface meltwater routing models, the daily minimum moulin input generally corresponds to daily maximum effective pressure (i.e. minimum water pressure) in these experiments. The minimum effective pressure (maximum water pressure) occurs several hours later as moulin inputs increase again into a subglacial system that has shut down due to ice creep in the absence of high meltwater input required to maintain a larger gap height. A magnified example of this timing is seen in Figure 5, which presents the average two-day cycle of moulin discharge input using the three routing models overlaid with effective pressure in IDC1 (results for the other three IDCs are presented in Figures S8-S10). All three routing models achieve minimum moulin discharges around 09:00-11:00 and minimum effective pressures around 17:00-19:00, yielding a time lag of 8-9 hours. In contrast, the RCM instantaneous runoff achieves minimum moulin discharge around 00:00 and minimum effective pressure around 10:00. The timing of effective pressure produced using RCM instantaneous runoff is visibly different than with the routing methods. Interestingly, the timing of minimum effective pressure simulated by the RCM instantaneous runoff is very close to that of maximum effective pressure simulated by the routing models (~ 1 hour).

In general, we find that while the moulin inputs generated by the different surface routing models produce variations in diurnal range of effective pressure, the qualitative channelization behaviour and temporal mean effective pressure over the 31-day simulations are relatively consistent between models (Figures 3 and 5). Note that the small, idealized subglacial model domain was designed to compare the impact of each supraglacial routing model on local subglacial hydrology behaviour in the vicinity of a moulin input at the bed. These numerical experiments explore differences resulting from forcing by the different surface routing methods under the same conditions and are explicitly not a representation of realistic subglacial evolution in a given catchment. We acknowledge that uncertain parameters and the particular boundary conditions certainly play a role in the specific behaviour and pressure magnitudes shown here, and these results are intended solely to illustrate and compare differences between methods in an idealized setting. To investigate the influence of the upstream boundary condition in our domain, we have compared sensitivity experiments (not shown) using a steady moulin input of 15 m$^3$/s (the mean input of the SUH routing method), run to a steady subglacial drainage configuration, using a no-flux ($dh/dx=0$) Neumann boundary as used in the experiments presented here vs. a Dirichlet boundary, setting the head equivalent to 50% of the ice overburden pressure (identical to the downstream boundary condition). The spatial mean effective pressure (our quantity of interest) in the case with Dirichlet conditions at both upstream and downstream edges is 5% higher than with the Neumann upstream condition. The prescribed pressure of the downstream condition also influences the magnitude of effective pressure: setting this to 40% of overburden results in a mean effective pressure 17% higher than with the 50% overburden condition; increasing the downstream boundary to 60% overburden decreases the resulting mean effective pressure by 15%. We find that imposing high-pressure boundary conditions of 90% overburden at the upstream and downstream edges appears to somewhat inhibit the development of the subglacial system in response to the moulin inputs as the entire domain is highly pressurized (and is therefore not as helpful for our purpose of comparing the behaviour with inputs derived from the different routing models).

As such, our results should not necessarily be extrapolated to infer large-scale or seasonal evolution of the subglacial hydrologic system in response to different surface forcings; however, the results do provide insight into the potential diurnal sensitivity of the subglacial system to changes in supraglacial meltwater routing and the associated modification of the discharge hydrograph. For example, the amplitude of diurnal effective pressure variation for a particular day may range from < 0.5 MPa to > 3.0 MPa in this experimental setup, depending on the surface method used (Figure 3). We hope this will serve as inspiration for future work to pursue broader-scope simulations to thoroughly investigate the larger- and longer-scale effects of surface input variations on effective pressure and ice dynamics.

## 3.3 Temporal evolution of moulin discharge simulations

To illustrate the strong potential dynamism of supraglacial stream/river network extent on bare ice, Figure 6 presents dynamic supraglacial stream/river networks of IDC1 under different channel initial $A_c$ thresholds (proxy for time). Numerous small supraglacial streams are generated when $A_c$ is set to 5000 m$^2$ (Figure 6a), while only large main stem channels are

obtained when $A_c$ is set to $10^6$ m$^2$ (Figure 6d), yielding drainage density varying from 19.3 m$^{-1}$ to 0.8 m$^{-1}$. Such strong, seasonal variation in supraglacial stream/river network extent is well-supported by previous high-resolution remote sensing studies, which report similar expansion and contraction of actively flowing supraglacial stream/river networks during the melt season (Smith et al., 2015; Smith et al., 2017; Yang et al., 2018).

Well-developed supraglacial stream/river networks (e.g., simulated using $A_c = 100$ m$^2$) route surface meltwater efficiently (Smith et al., 2015), yielding high UH and moulin discharge peaks, whereas relatively poorly-developed supraglacial stream/river networks indicate relatively slow routing of meltwater in expanding interfluve zones (Yang et al., 2018), yielding smaller and delayed UH and moulin discharge peaks (Figures 7a and 7b). As such, temporal evolution of supraglacial stream/river networks substantially alter surface meltwater routing processes and yield dynamic unit

hydrographs and moulin discharges. This result is consistent with Montgomery and Foufoula-Georgiou (1993) and van Meerveld et al. (2019), which found that river network extent controls the response of catchments to precipitation (melt in our case). During the first 10 days (240 hours), the channel initiation thresholds $A_c$ are small so numerous supraglacial streams/rivers are developed; thereby, the dynamic $A_c$ RWF routing performs similarly with SUH, 2 m SRLF, and the original RWF. As supraglacial stream/river networks shrink (i.e., $A_c$ becomes larger), our simple modelling experiment (by

increasing $A_c$ thresholds every 5 days for the month of July 2015) suggests a gradual dampening of diurnal variations until, by the end of the month, such variations are ~50 % smaller than the same routing method using a smaller $A_c$ value (Figure 7c). Moreover, in the dynamic $A_c$ RWF routing, the dampening effect displayed in the moulin discharge inputs carries over into the resulting effective pressure cycles as well (Figure 7c).

### 3.4 Effects of DEM spatial resolution on surface meltwater routing

RWF routing is largely unaffected by DEM spatial resolution, but SRLF routing is significantly affected. Resampling 2 m ArcticDEM data to 5 m and 10 m yields similar RWF UHs and moulin discharges (Figures 2e and 2f), but progressively lower amplitude SRLF UHs (Figures 2c and 2d). RWF relies on high-resolution DEMs to calculate interfluve transport distance and cannot use DEM of coarser resolution (>10 m) (Yang et al., 2018). At 90 m resolution, SRLF UH shapes exhibit diminished and delayed peaks (Figure 2c). For IDC1, for example, use of a 90 m DEM yields a ~0.10 UH peak at hour 9 and a ~0.04 UH peak at hour 17, whereas 2 m DEM yield a ~0.15 UH peak at hour 5 with all meltwater evacuated within 15 hours.

     SRLF applied on higher-resolution DEMs yields larger peak moulin discharges and diurnal discharge signals (Figure 2d). For IDC1, SRLF-routed peak moulin discharge derived from 2 m DEM is 52.4 % larger than that from 90 m DEM, while the corresponding value for SRLF-routed diurnal discharge range is 179.0 %. This finding indicates that SRLF

performance is strongly controlled by DEM spatial resolution, with coarser resolutions resulting increased damping of simulated moulin hydrographs. The dampening effect of SRLF moulin discharge induced by DEM spatial resolutions carries

over into the resulting effective pressure cycles as well (Figure 3b), urging caution when using coarse-resolution DEMs to route surface meltwater and to simulate subglacial effective pressure.

## 4 Discussion

### 4.1 Inter-comparison of surface meltwater routing models

This study conducts an inter-comparison of three surface meltwater routing models (SUH, SRLF, and RWF). Due to the lack of field-measured moulin hydrographs, it is difficult to determine which surface meltwater routing model performs most realistically over large areas and long times. However, our simulations of moulin discharges (Figures 2, and S1-S3) can provide some insightful information to qualitatively evaluate the performance of certain routing methods. First, inclusion of surface meltwater routing introduces lower peak moulin discharges and delayed time to peak relative to the RCM

instantaneous runoff. Second, for IDCs with similar areas and elevations as those examined here, simulated peak moulin discharge consistently occurs between 19:30 and 22:00 (Chandler et al., 2013; Smith et al., 2017). The timing of peak discharge suggests that several routing scenarios are unlikely to be realistic, including any technique that uses unmodified RCM outputs as these values qualitatively underestimate peak discharge time.

      Moreover, SUH and RWF both rely on several empirical parameters ($C_p$ and $C_t$ for SUH, and $v_h$ and $v_c$ for RWF)

calibrated from a moulin hydrograph measured at the Rio Behar catchment, southwestern GrIS during a very short time period (72 hours), July 2015 (Smith et al., 2017). SRLF is applicable over large spaces and long times because it only relies on DEM to calculate meltwater flow velocities (Banwell et al., 2012). In this study, we assume these empirical parameters are transferable over space and time but this assumption needs further validation. It may hold for ice sheet surface with similar hydrologic and glaciological environments but is problematic to apply over large spaces and long times due to

evolving ice surface characteristics. Multiple independent, long-term moulin hydrographs will help eliminate the need for this assumption.

### 4.2 Influence on diurnal subglacial pressure variations

      Our results demonstrate that the routing models and DEM resolution can modulate the diurnal variability of both surface meltwater transport and associated moulin inputs (Figure 2). Surface meltwater routing alters hourly inputs, with

only slight changes to total integrated daily moulin input. However, hourly variations in moulin discharge can influence ice dynamics (Figure 3). Diurnal variations in subglacial pressure, for example, can trigger variations in ice velocity and induce additional surface-to-bed connections (Carmichael et al., 2015; Hoffman et al., 2018). The diurnal amplitude of moulin discharge also affects the transient evolution of the subglacial drainage system without sustained input to maintain efficient drainage channels, effective pressure decreases, resulting in increased sliding velocities, potentially influencing seasonal ice

displacement (Hewitt, 2013; van de Wal et al., 2015). As such, including the diurnal signal of moulin discharge may be

important for accurately modelling surface-to-bed meltwater connections (Figures 3-5 and S4-S10). As we see reflected in both the magnitude of effective pressure diurnal amplitudes (Figure 3) and in the timing of minimum/maximum effective pressure relative to moulin inputs (Figure 5), different surface routing methods yield a range of diurnal behaviour. However, as described in Section 3.2, the temporal mean effective pressure produced in our subglacial hydrology simulations with inputs from the various routing models is relatively consistent between routing models even though the amplitude varies greatly. This suggests that depending on the time scale of interest, diurnal inputs may not actually be vital to capture the relevant ice dynamics, and this should be investigated in future work with detailed catchment-scale (or larger) modelling.

Many subglacial hydrology models commonly invoke a numerical term (the "englacial void ratio") to represent englacial storage in order to provide short term storage and release of meltwater that cannot be accommodated rapidly within the subglacial system, in the absence of more realistic representation of supraglacial and englacial storage (Hewitt, 2013; Werder et al., 2013; Hoffman et al., 2016). In SHAKTI, this englacial void ratio is also included as an option (Sommers et al., 2018), but our simulations considered here do not employ this term in the equations. Our results present that the supraglacial hydrologic system acts as short-term storage for surface-derived meltwater, as exhibited by the time lag of moulin inputs between models; therefore, application of an appropriate surface meltwater routing scheme may reduce the dependence of some subglacial models on a somewhat arbitrary englacial storage term to produce realistic diurnal effective pressure variations and timing lags (Werder et al., 2013; Hoffman et al., 2016). It is conceivable, for example, that routing models could help to parameterise the storage term built into subglacial hydrology models; or that some portion of this term should in fact be apportioned to supraglacial routing delays. The assumption of surface inputs being instantaneously delivered to the subglacial system is an approximation that ignores the largely uncertain complex flow paths through the englacial system from surface to bed, but it is reasonable to assume that variability in surface inputs should influence variability in inputs to the subglacial system.

Supraglacial routing delays can affect the amplitude and timing of subglacial pressure variations (Figures 3-5 and and S4-S10). While there is limited evidence that within a fully coupled ice dynamics/subglacial hydrology model that diurnal variations in effective pressure can contribute to changes in ice dynamics (Hewitt, 2013), the evolution of the supraglacial hydrologic system can substantially alter effective pressure on diurnal timescales (e.g. Figure 7c). Subglacial channel development is key to terminating early melt season accelerated sliding (Hoffman and Price, 2014) and subglacial channelization occurs more readily and persistently under constant supraglacial meltwater inputs (Schoof, 2010; Poinar et al., 2019). The appropriate choice of a time evolving surface meltwater routing scheme may, in addition to providing the best representation of diurnal effective pressure variations, result in an improved representation of subglacial evolution and improve quantitative representation of seasonal variations in ice flow in coupled models. Our results, however, show that while meltwater inputs with highly variable diurnal amplitudes yield effective pressures with highly variable diurnal amplitudes, the channelization behaviour and temporal mean effective pressure over the 31-day simulations are quite similar across routing models (Figure 5). Further work is needed to explore the implications of using sub-daily inputs versus time-averaged inputs on long-term subglacial hydrology and ice dynamics in realistic, large domains. The case may be that the

peaks and troughs of large-amplitude diurnal input variations effectively balance out to yield equivalent results as time-averaged inputs.

## 4.3 Influence of seasonal supraglacial drainage evolution on meltwater routing

Seasonal evolution of ice surface drainage pattern can substantially alter surface meltwater routing and moulin discharge characteristics (Figures 2 and S1-S3). Such changes also involve the removal and deposition of a winter snowpack (Nienow et al., 2017) and the development of a weathering crust on base ice (Cooper et al., 2018), both of which can reduce the prevalence of open-channel flow (Yang et al., 2018). While these processes cannot be explicitly represented with the meltwater routing parameterizations described here, time evolving UHs can integrate the impact of seasonal changes in the relative proportions of porous and open-channel flow. As such, the relative ease of creating dynamic UHs to mimic seasonal evolution of supraglacial stream/river networks is a distinctive advantage of RWF (Figures 7a and 7b).

The development of time varying UHs provides an opportunity to simulate dynamic moulin discharges (Smith et al., 2017; Yang et al., 2018). Such time-varying UHs could be important to realistically modelling the evolution of the subglacial drainage system and the development of subglacial channels – a limited supraglacial stream/river extent will dampen the diurnal range of meltwater inputs into the subglacial system and maintain a more constant subglacial pressure (e.g. Figure 7c), which in turn results in less variation in ice motion. Furthermore, limiting diurnal variations may result in more rapid growth of subglacial channels early in the melt season (Schoof, 2010; Hewitt, 2013), thus, potentially produce a more realistic transition to primarily channelized drainage (Banwell et al., 2013; Banwell et al., 2016).

## 4.4 Impact of DEM resolution on supraglacial meltwater routing

Analysing the influence of DEM spatial resolution on the hydrological response of catchment to precipitation is an important research topic in terrestrial hydrology (Zhang and Montgomery, 1994; Hancock et al., 2006). We expand it to ice sheet hydrology and attempt to investigate the impact of DEM spatial resolution on the hydrological response of supraglacial catchment to surface melt. We found that RWF routing is not affected by DEM spatial resolution, whereas SRLF routing is (Table 2, Figures 2 and S1-S3). Notably, RWF can only operate on high resolution (<10 m) DEMs, whereas SRLF can be applied to coarser resolution DEM as well. The 5 m SRLF UHs and resultant moulin discharges best match with RWF and SUH simulations (Figures 2 and S1-S3). This is consistent with the finding that "the most appropriate DEM grid size for topographically driven hydrologic models is somewhat finer than the interfluve scale identifiable in the field" (Zhang and Montgomery, 1994). This behaviour occurs because a coarser-resolution DEM represents bare ice surface topography more smoothly and yields lower bare ice surface slopes, similar to terrestrial topography (Zhang and Montgomery, 1994); lower topographical slopes yield smaller velocities via Manning's open-channel equation.

Different meltwater routing velocities control the meltwater transport time because meltwater flow length is minimally impacted by DEM resolutions. As a result, lower SRLF meltwater routing velocities induce longer meltwater transport time

and distribute diurnal surface runoff more smoothly over time (Figures 2c and 2d), whereas stable flow length contributes to RWF's better performances under different DEM resolutions (Figures 2e and 2f).

## 4.5 Future research directions of surface-to-bed meltwater connection

A challenge remaining unsolved is to map or simulate seasonal evolution of supraglacial stream/river networks. Spatial extent of supraglacial stream/river networks determine partition of interfluve and open-channel zones and thereby controls IDC hydrologic response to surface melt. The dynamic $A_c$ thresholds provide a simple simulation of supraglacial stream/river network evolution but may not represent the realistic evolution (Yang et al., 2018). In future, a more solid relationship between RCM surface runoff and supraglacial drainage pattern should be built.

Moreover, all three routing models described in this study have limited capacity to represent complex processes of ice surface degradation ("weathering crust") from penetration of solar radiation and associated subsurface melting (Karlstrom et al., 2014; Cooper et al., 2018). SUH relies on catchment and stream/river network drainage pattern (Smith et al., 2017) and does not consider the possibility of shallow subsurface flow through weathering crust. While the snow component of SRLF can handle seasonal snow cover, its bare ice component also cannot accommodate a permeable bare ice surface (Arnold et al., 1998; Banwell et al., 2012). RWF, in contrast, can accommodate shallow subsurface flow through weathering crust using field-calibrated interfluve flow velocities, but does not support a third option of seasonal snow cover (Yang et al., 2018). As a result, RWF cannot be used to simulate moulin discharge for snow-covered IDCs during the early melt season. We suggest that future work should pursue an appropriate parameterization of shallow porous media flow through both seasonal snow cover and weathering crust, in order to accurately describe meltwater transport in this area throughout the duration of the melt season.

Finally, the subglacial hydrology simulations in this study should be interpreted as an experimental demonstration of how different meltwater routing methods influence local subglacial hydrology in the vicinity of a moulin through variations in moulin inputs. As with other models, the specific effective pressures presented here are admittedly influenced by uncertain modelling parameters and boundary conditions, and should be viewed as a comparison between methods, not as a representation of real behaviour at the bed in these catchments. Regional simulations covering multiple IDCs in a fully coupled ice dynamics-subglacial hydrology model remain for future work, an important next step in understanding the surface-to-bed meltwater influence on regional and large-scale ice dynamics. A more direct comparison of model results with observations will be appropriate for this future work, in which the entire drainage catchments are modelled with multiple moulin distributed in areas where borehole observations are available or relevant quantities may be inferred from radar products (Chu et al., 2016).

## 5 Conclusions

Surface meltwater routing is crucial for understanding the surface-to-bed meltwater connection of the Greenland Ice Sheet but remains poorly studied to date. This study presents a first inter-comparison of three different meltwater routing models and employs a subglacial hydrologic model to explore the impacts of their differences on subglacial effective pressure (ice overburden pressure – water pressure) in the vicinity of a moulin in an idealized subglacial setting. Results show that inclusion of surface meltwater routing introduces significantly small peak moulin discharges and delayed time to peak relative to the RCM instantaneous runoff. Different surface meltwater routing models, as well as different spatial-resolution DEMs and seasonal evolution of the supraglacial stream/river networks, induce variable diurnal moulin discharges and effective pressures, which in reality would influence ice sliding velocity. While the different routing models produce different diurnal amplitudes in effective pressure variation, the qualitative channelization behaviour and temporal mean effective pressures are relatively consistent across surface meltwater routing models. Together, these findings urge caution for better representations of surface meltwater routing and moulin discharge simulations to drive subglacial hydrology and ice dynamics models, as well as highlight the need for further research to investigate the cumulative effects of diurnal inputs to the subglacial drainage system and the relevant impacts on ice dynamics through detailed, large-scale modelling.

**Data availability**

The ArcticDEM data are available at the Public HTTP Data Repository of Polar Geospatial Center at the University of Minnesota (https://www.pgc.umn.edu/data/arcticdem/). Hourly MAR (Modèle Atmosphérique Régionale) 3.6 model data can be accessed by contacting Xavier Fettweis (xavier.fettweis@uliege.be). Codes used to generate surface routing are available via https://doi.org/10.6084/m9.figshare.11635932.v1. The SHAKTI subglacial hydrology model is freely available as part of the Ice Sheet System Model (https://issm.jpl.nasa.gov/).

**Author contributions**

KY and AS designed the study. KY and AS performed the data analysis. KY wrote the paper with contributions from all authors.

**Competing interests**

The authors declare that they have no conflict of interest.

**Acknowledgements**

Kang Yang acknowledges support from the National Key R&D Program (2018YFC1406101), the National Natural Science Foundation of China (41871327), and the Fundamental Research Funds for the Central Universities (14380070). Laurence C. Smith acknowledges the support of the NASA Cryosphere Program (80NSSC19K0942) managed by Dr. Thorsten Markus.

5    ArcticDEMs are provided by the Polar Geospatial Center at the University of Minnesota under NSF-OPP awards 1043681, 1559691, and 1542736.

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

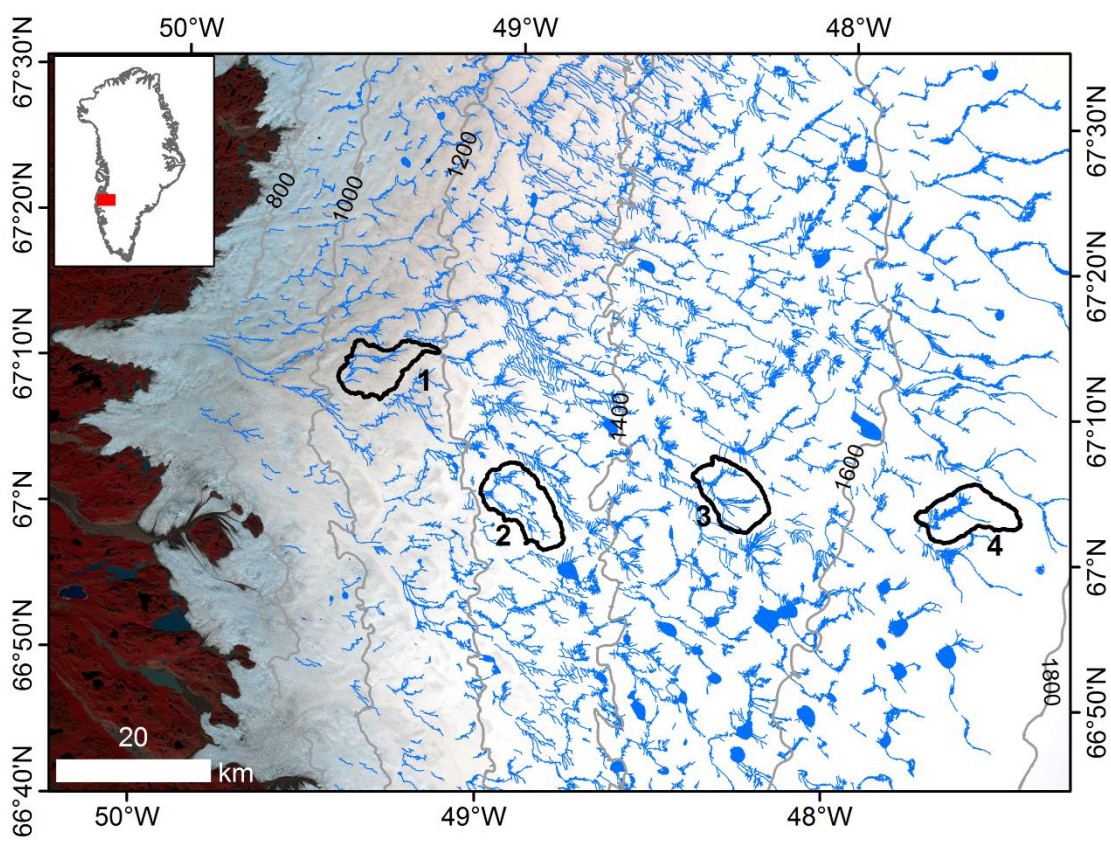

**Figure 1. Four supraglacial internally drained catchments (IDCs) on the southwest Greenland Ice Sheet were selected for inter-comparison of surface meltwater routing models. IDC topographic boundaries (black lines) are extracted from high-resolution (2 m) ArcticDEMs. Supraglacial river networks and lakes (blue lines) are mapped from a Landsat-8 panchromatic image (Yang and Smith, 2016). Base map is 10 m Sentinel-2 image acquired on 30 July 2018.**

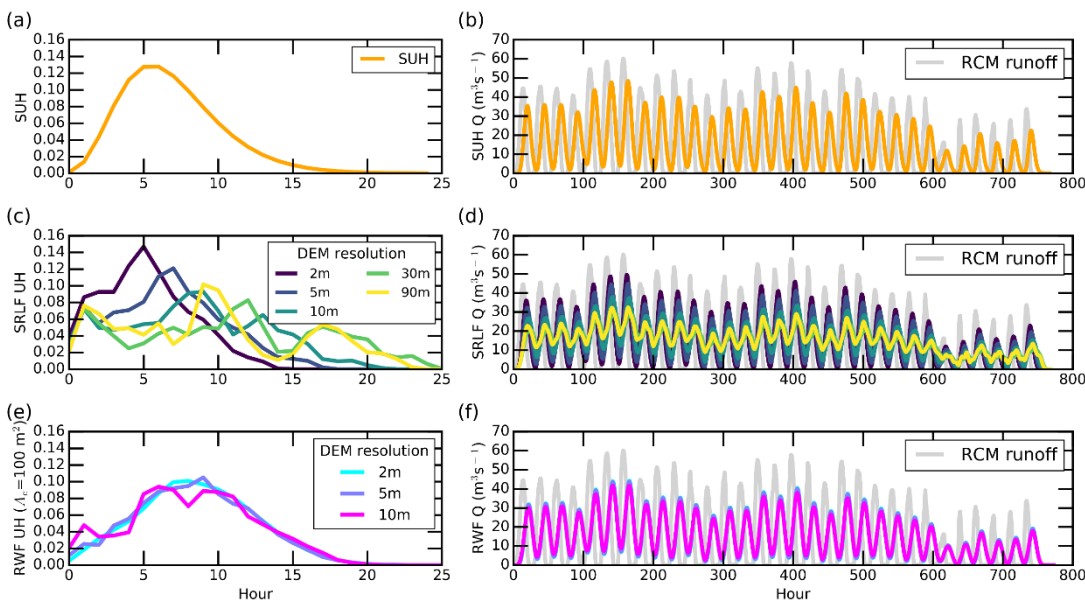

**Figure 2. Presentation of Unit Hydrographs (UHs) (column 1) and moulin discharges (column 2) of IDC1 during July 2015, as simulated by three surface meltwater routing models (SUH, RWF, and SRLF). Simultaneous RCM runoff (grey line) is shown to indicate the effect of surface meltwater routing process on moulin discharge.**

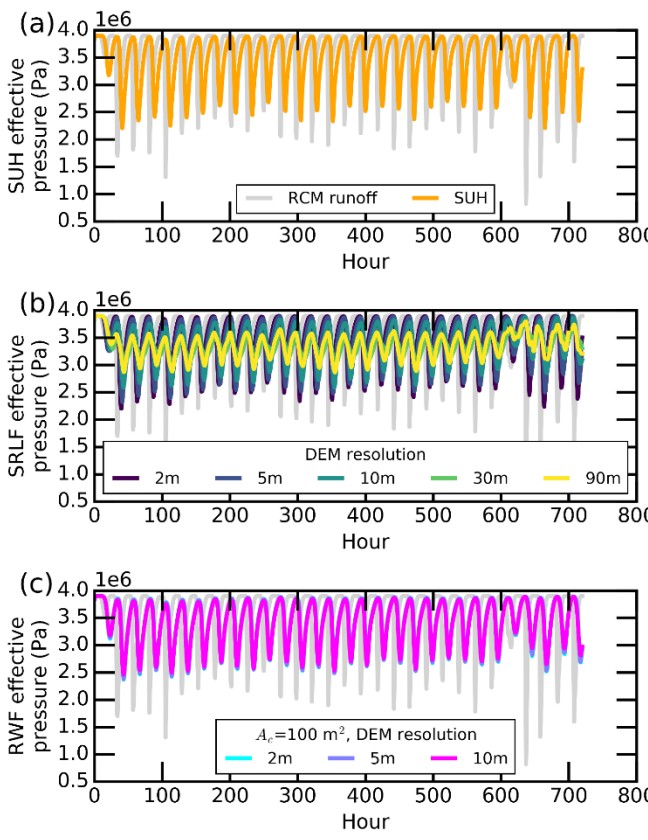

**Figure 3. Subglacial effective pressures for IDC1 as simulated by SHAKTI, with inputs to the subglacial system via a single moulin prescribed by the moulin discharges (shown in Figure 2) generated by the various surface meltwater routing models. The effective pressure shown is the spatial mean over the entire 1-km square domain which contains the moulin input at its center.**

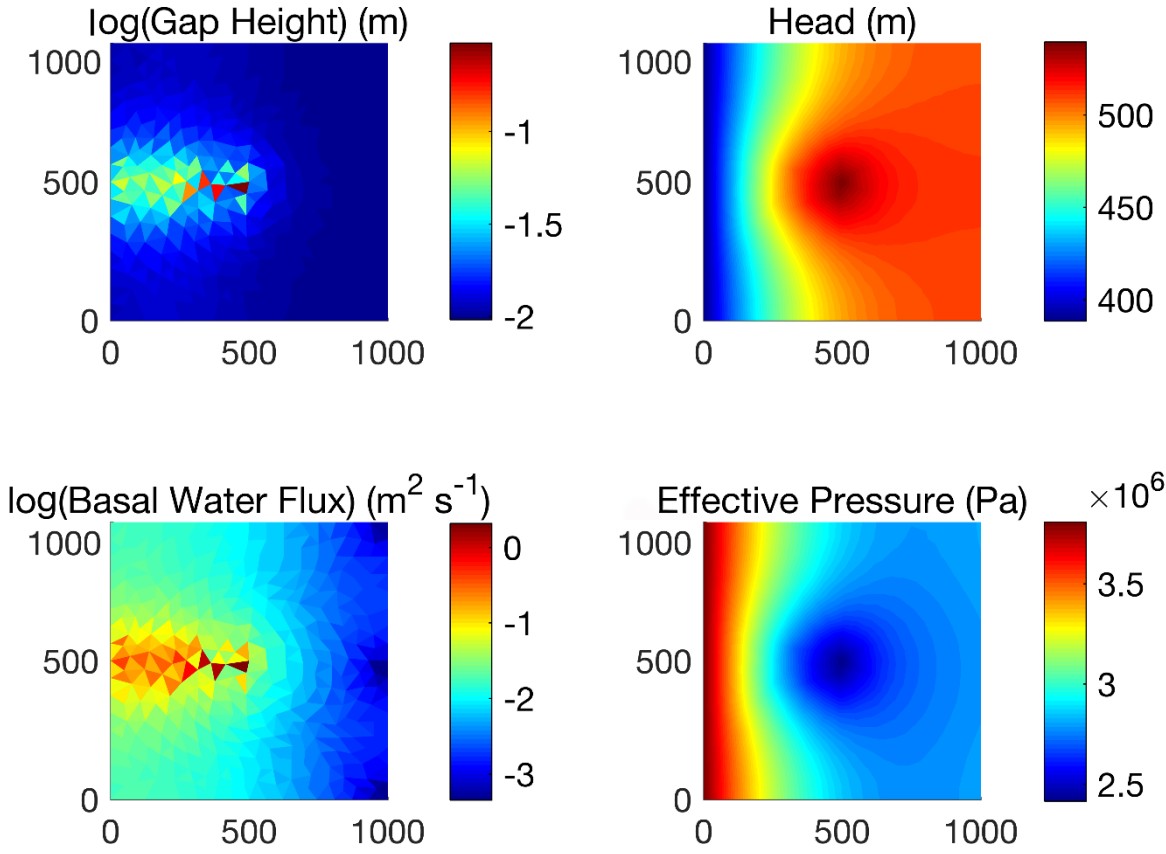

**Figure 4. Snapshots of dynamic subglacial hydrology fields on day 23 in IDC1 using the SUH routing method to drive moulin input (see full animation of channel evolution and fluctuation in the supplement). An efficient channelized drainage pathway develops from the moulin location at the center of the domain to the outflow at the left, characterized by higher gap height, water flux, effective pressure, and lower hydraulic head than its surroundings perpendicular to flow.**

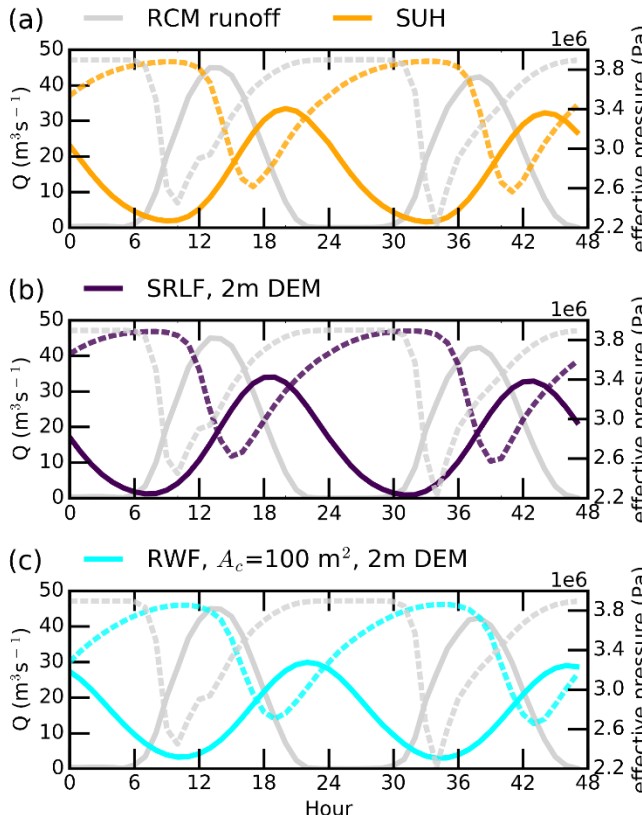

**Figure 5. The average two-day cycle of moulin discharge (*Q*) for IDC1 during July 2015 derived from Figures 2 and 3. The daily minimum input in supraglacial moulin discharge (solid lines) corresponds generally to maximum effective pressure (dashed lines), and is followed within 8-9 hours by the daily minimum effective pressure (maximum subglacial water pressure). This suggests that the system shuts down due to creep with low meltwater input, and becomes highly pressurized as meltwater input increases again. As the new water inputs are accommodated, efficient pathways reform and effective pressure increases (subglacial water pressure decreases).**

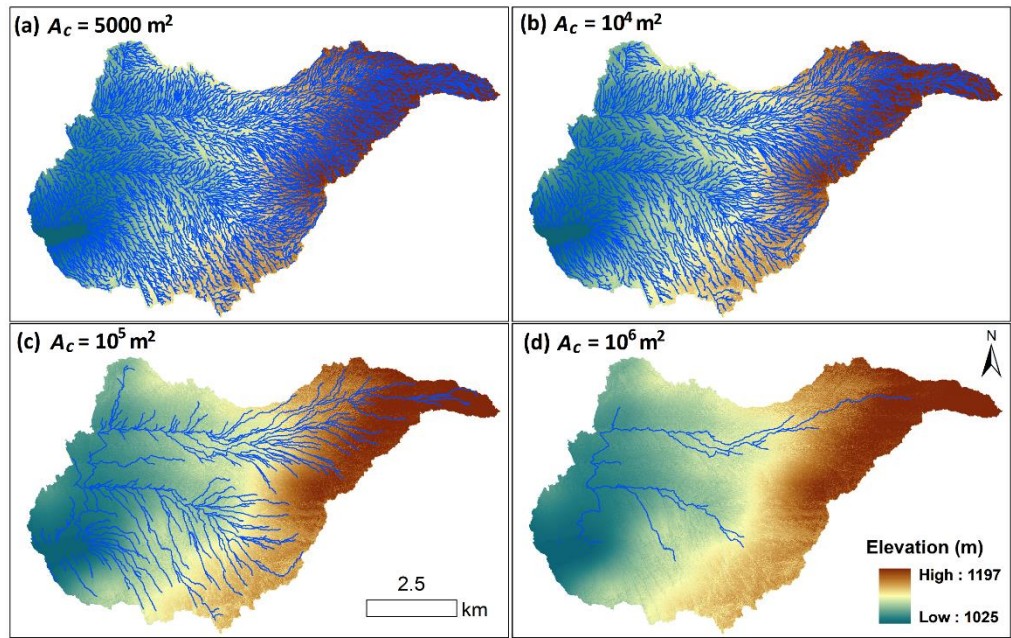

**Figure 6. Variable supraglacial stream/river network for IDC1, as simulated by applying variable accumulative area threshold (channel initial threshold, $A_c$) values to ArcticDEM.**

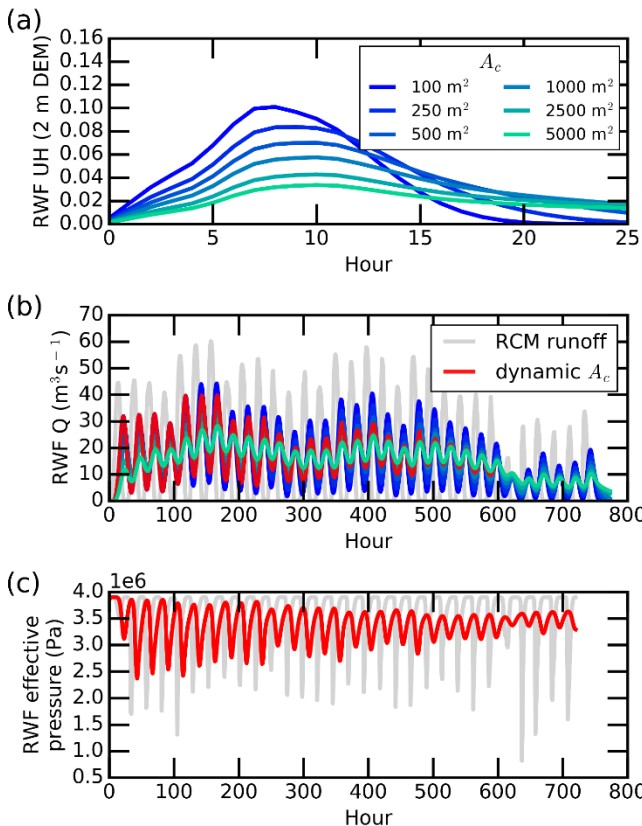

**Figure 7. Presentation of (a) Unit Hydrographs (UHs), (b) moulin discharges, and (c) subglacial effective pressure of IDC1 during July 2015, as simulated by dynamic $A_c$ values. $A_c$ is the cumulative contributing area required to initiate a supraglacial meltwater channel and dynamic $A_c$ values are used as proxy for time to simulate the temporal evolution of supraglacial stream/river networks. Simultaneous RCM runoff (grey line) is shown to indicate the effect of surface meltwater routing process on moulin discharge.**

**Table 1. A brief summarization of surface meltwater routing models.**

| Model | Meltwater Routing | Applicable on bare ice surfaces | | Applicable on snow surfaces | Parameter dependency | DEM dependency | Case study |
|---|---|---|---|---|---|---|---|
| | | Interfluve | Open-Channel | | | | |
| **Instantaneous RCM runoff** | No | - | - | Yes | No | No | (McGrath et al., 2011; Bartholomew et al., 2012; Rennermalm et al., 2013; Fitzpatrick et al., 2014) |
| **Snyder Synthetic Unit Hydrograph (SUH)** | Yes | - | - | Yes, but model parameters need recalibration | $C_p$, $C_t$ are calibrated using a field-measured moulin hydrograph | No | (Smith et al., 2017) |
| **Surface Routing and Lake Filling (SRLF)** | Yes | No | Yes | Yes | No | DEM is required to calculate meltwater flow velocities for all catchment cells | (Arnold et al., 1998; Willis et al., 2002; Banwell et al., 2012; Banwell et al., 2013; Arnold et al., 2014; Banwell et al., 2016; de Fleurian et al., 2016; Koziol and Arnold, 2018) |
| **Rescaled Width Function (RWF)** | Yes | Yes | Yes | Yes, but model parameters should be recalibrated | Interfluve and open-channel flow velocities ($v_h$ and $v_c$) are calibrated using a field-measured moulin hydrograph | High-resolution (<10 m) DEM is required to calculate interfluve flow path length | (Yang et al., 2018) |

**Table 2. Summary of four study catchments.**

| Catchment ID | | IDC1 | IDC2 | IDC3 | IDC4 |
|---|---|---|---|---|---|
| Area (km²) | | 53.0 | 66.9 | 57.3 | 58.5 |
| Mean Elevation (m) | | 1054 | 1248 | 1473 | 1646 |
| Mean surface slope (m/m) | | 0.018 | 0.020 | 0.008 | 0.008 |
| Mean bed elevation (m) | | 207 | 309 | 247 | 222 |
| Mean bed slope (m/m) | | 0.050 | 0.075 | 0.036 | 0.022 |
| Mean ice thickness (m) | | 847 | 939 | 1226 | 1424 |
| Mean ice flow velocity (m/a) | | 116 | 99 | 98 | 73 |
| Distance to ice edge (km) | | 25 | 40 | 70 | 100 |
| Peak discharge time | RCM | 13-15 | 13-15 | 13-14 | 13-14 |
| | SUH | 19-20 | 20-21 | 19-20 | 19-20 |
| | SRLF 2m, 5m, 10m, 30m, 90m | 18-19, 20-21, 21-22, 22-23, 22-23 | 18-19, 19-20, 21-22, 21-22, 21-22 | 17-18, 18-19, 20-21, 22-23, 22-23 | 17-18, 19-20, 23, 23, 23 |
| | RWF | 22-23 | 21-22 | 20-21 | 21-22 |

**Table 3. Constants and parameters used in subglacial hydrology simulations.**

| Symbol | Value | Units | Description |
|--------|-------|-------|-------------|
| $\rho_w$ | 1000 | kg m$^{-3}$ | Bulk density of water |
| $\rho_i$ | 910 | kg m$^{-3}$ | Bulk density of ice |
| A | $5 \times 10^{-25}$ | Pa$^{-3}$ s$^{-1}$ | Ice flow-law parameter |
| n | 3 | Dimensionless | Ice flow-law exponent |
| $b_r$ | 0.1 | m | Typical height of bed bumps |
| $l_r$ | 2.0 | m | Typical spacing between bed bumps |
| g | 9.8 | m s$^{-2}$ | Gravitational acceleration |
| $\omega$ | 0.001 | Dimensionless | Parameter controlling nonlinear transition between laminar and turbulent flow |
| L | $3.34 \times 10^5$ | J kg$^{-1}$ | Latent heat of fusion of water |
| G | 0.05 | W m$^{-2}$ | Geothermal flux |
| $c_t$ | $7.5 \times 10^{-8}$ | K Pa$^{-1}$ | Change of pressure melting point with temperature |
| $c_w$ | $4.22 \times 10^3$ | J kg$^{-1}$ K$^{-1}$ | Heat capacity of water |
| $\upsilon$ | $1.787 \times 10^{-6}$ | m$^2$ s$^{-1}$ | Kinematic viscosity of water |
| $e_v$ | 0 | Dimensionless | Englacial void ratio |