# Peer review of "Inter-comparison of surface meltwater routing models for the Greenland Ice Sheet and influence on subglacial effective pressures"

_The Cryosphere, 2019_

## Referee Comment (RC1) · Basile de Fleurian (Referee) · 6 Dec 2019

**Review of the article entitled "Inter-comparison of surface meltwater routing models for the Greenland Ice Sheet and influence on subglacial effective pressures."**

**1 General comments**

This study focuses on different ways of treating the supraglacial drainage of water at the surface of ice sheets. The region of interest studied here is Russell glacier in South West Greenland. The authors presents an intercomparison of three different surface routing models and compare their results to the output of a Regional Climate Model (RCM). The different inputs are further compared through by using them as the forcing quantity provided to a subglacial drainage model. The conclusion of the study are that the use of a supraglacial drainage system allows to get a better representations of the lag of the water input to the moulins. Some sensitivity among models also allow to quantify the impact of the Digital Elevation Model resolution on the drainage characteristics

This study provides an interesting insight into the differences that arise from the use of different supraglacial drainage model. However, my impression is that this study should be further refined in order to be more understandable and provide a usable tool for the community. I find that the presentation of the different models and their results is lacking detail and clarity. Moreover I am concerned by the choice that were made with regards to the boundary condition that are applied to the subglacial hydrology model. Details of my concerns and potential improvement are given bellow section by section.

**1.1 Abstract**

The abstract is quite hard to read due to the accumulation of acronyms. Where possible I would urge the authors to refrain from using acronyms in this part of the papers. It might be beneficial to simplify the abstract to make it more accessible to readers which might afterwards gather the details of the study in the rest of the paper. As an example, the author could state that they compare three surface meltwater routing models at this point

without specifying those models, the list of variables line 21 (page 1) could be omitted and replaced by "key variables".

**1.2 Introduction**

The introduction gives a succinct outlook on the motivations of the study. This could be developed further to point out the current lack of representation of the supraglacial drainage system and the necessity to have a better representation of this system. The description starting on line 20 (page 2) would fit better in a method section of the paper. Moreover, some terms defined in the introduction (such as Unit Hydrograph or Internally Drained Catchments) might not be familiar to the Cryosphere community and the author should consider defining those in more details.

I don't completely agree with the statement starting on line 10 (page 2) to my knowledge supraglacial meltwater routing is usually simplified in subglacial hydrology models (*e.g.* Banwell et al., 2016; de Fleurian et al., 2016) it would be interesting to have some citation here that present studies directly using an RCM as their water input. I am not sure that the citation to Flowers et al. (2018) is relevant in this context or I missed the point of the author here. Further down, the citation to Bartholomew et al. (2011) seemed to be misplaced here as this specific study treats about observations rather than modelling.

**1.3 Study area and data source**

This section is missing a major information as the study area is actually never named. The Russell glacier region will be familiar to most of the reader interested of the subject but mention of it should still appear in the paper. My opinion is that this section should be merged into a section 3 (Methods and Data). Regarding the content of the present section it is not clear to me how and why the IDCs that are presented in this section were generated and why those specific IDCs have been chosen.

**1.4 Methods**

The description of the different models here is quite brief and some more details could be provided. Particularly it would be interesting to have a better overview of the advantages and drawbacks of each models. The paragraph starting line 15 (page 5) would fit better in the introduction of the study rather than here. Subsections 3.5 and 3.6 refer to the sensitivity studies that where performed for some of the model, it could be beneficial to transfer those sections into the descriptions of the relevant models. That would outline the advantages and potential drawbacks of the models and would clarify the overall setup of the experiments.

From the references that were provided in the paper regarding the SRLF model I understand that this model is routing water with different equations if it sits on snow

or on bare ice. From the model description given here it seems that only the bare ice formulation was used. Is that so? If yes the reasons for this choice should be explained.

As stated above my main concern with this study is the way in which the subglacial hydrology model is set-up. In my opinion the boundary condition that is given for the left edge of the domain is not realistic, I do not think that we expect to find water pressure at the atmospheric pressure anywhere under the ice sheet. A more sensible choice would be to set the water pressure at a given fraction of the overburden pressure. A change of boundary condition would need to perform new simulations but I would expect a good argumentation on the choice of the present boundary condition if it is to be kept. I also do not understand why the slopes of the bed and surface, and velocities are not taken from the values of the IDCs as is done for the ice thickness. As it stands now I have a hard time trusting the results from the subglacial hydrology model as it seems that the downstream boundary that is currently set is exerting an important control on the whole domain. I would also note that the Figure 2 related to this section is not very informative and could probably be omitted.

**1.5   Results**

In general I find the presentation of the results quite hard to follow. This might come from the structure that was chosen by the author, from the presentation of the figures or both. I also wonder why only the results from IDC 1 are presented, it appears from the supplementary figures that the results from the four IDCs are quite similar but this should be stated. I also expect that changing the boundary conditions and parameters of the subglacial hydrology model may alter those results.

Regarding the presentation of the results, it would be clearer to me if the author would describe first the results of the intercomparison itself before delving into the sensitivity studies that were performed on DEM resolution and the value of $A_c$. Comparing the results to a given reference might also help with the clarity of the text. I generally find the presented figures a bit too busy and so hard to read. Figure 3 is described a lot throughout the manuscript but the size of some panels make it hard to read. As for the text having specific figures for the intercomparison and the sensitivity study might help to lighten the figures. Figure 4, 6 and 7 however are not described in the Results part and should be included there. Lastly I have not seen any information with regard to the sampling of the effective pressure that is discussed, is it an average value or this value is taken at a specific point?

**1.6   Discussion and Conclusion**

The discussion of the manuscript is clear, it would however take advantage of the alterations suggested above for the Result section. Particularly describing all the figure in more details in the result section would help during the discussion. I also expect that the

changes required above regarding the subglacial hydrology model might have a significant impact on the results and should be taken into account in the discussion. I have noted a few minor concern on this section which are listed in the Specific comments bellow.

**2 Specific comments**

Bellow is a list of more specific comments throughout the manuscript given with line and page number:

- Page 1

    - Line 16: "ice surface" can be replaced by "ice sheet surface".
    - Line 17: "climatological melt " should be replaced by "surface metwater".
    - line 21: MAR abbreviation is not defined here.
    - Line 23: "input" can be replaced by "used as input"

- Page 2

    - Line 2: Surface melt is not restricted to the ablation zone but occur in the accumulation zone too.
    - Line 3: "Greenland ice surface" should be "Greenland ice sheet"
    - Line 3: "can be" should be "is"
    - line 14: Bartholomew et al. (2011) does not seem to be a fitting citation here as this paper treats of observations.

- Page 3

    - Line 3: "to discern", to is missing

- Page 4

    - Line 13: The parameters $C_p$ and $C_t$ should be explained.
    - Line 13: "time-to-peak in" reads strangely.
    - Line 15: I am not sure that the citations are needed here an interested reader will find those in Smith et al. (2017)

- Page 5

    - Equation 2: $t, t_c$ and $t_h$ are not described in the text.
    - Line 16: Replace "research" by "study".

- Page 6

  - Line 6: The contributing area ($A_c$) should be introduced and discussed in the model description.
  - Line 15: "compute" rather than "derive".
  - Line 19: "framework" could be omitted.

- Page 7

  - Line 6: "climate model" can be skipped here.
  - Line 7: The times given here do not agree with the one that are present on Figure 3. The author should chose which are the more relevant and keep them throughout.
  - Line 16: I don't agree with the statement on the smoothness of the UHs. From the figure it seems that the UHs from SUH are actually the smoothest of all.

- Page 7

  - Line 2: Shouldn't it be "potential dynamism"?

- Page 9

  - Line 16: Figure 7 actually shows the results from the three different models not only SUH. The comparison between the results of SHAKTI with the forcing from the RCM and the various models should be presented here to convince the reader of the advantage to use those models. As stated before, the set-up of the subglacial hydrology model should be corrected to give convincing results. I am also unsure of the location where the effective pressure presented on Figure 7 is sampled from the model.
  - Line 30: The study from Chandler et al. (2013) actually shows subglacial travel time. I don't see how this reference fits here.

- Page10

  - Line 18: Should be "bare ice".

- Figure 2: I don't think that Figure 2 is necessary and it could be skipped.

- Figure 3: This figure is quite hard to read as it holds a lot of information. I would suggest to plot on this figure only the optimal simulations for SRLF and RWF which would allow an easier and more fair intercomparison of the models. An other solution might be to split the figure to present the intercomparison on a specific figure and the sensitivity studies on others. Finally, a zoom on some relevant period for the

discharge and effective pressure would help the comparison of the different models. I also noticed a discrepancy here between the times given in the first column and the one of the text. It would be advantageous to introduce the RCM instantaneous runoff in the first column for ease of comparison.

- Figure 4: Figure four is barely described in the text, it should either be better described or completely omitted.

- Figure 5: $A_c$ is given here in km$^2$, it should be given in m$^2$ for consistency with the rest of the manuscript. The caption here could be shortened to its descriptive part.

- Figure 7: As for Figure 3 this figure is quite busy and should be simplified. The caption here is not adequate with some description missing and some discussion points that could be stripped.

- References: dois are missing from the references

**References**

Banwell, A., Hewitt, I., Willis, I., & Arnold, N. (2016). Moulin density controls drainage development beneath the greenland ice sheet. *Journal of Geophysical Research: Earth Surface*, *121*(12), 2248–2269. doi: 10.1002/2015JF003801

Bartholomew, I. D., Nienow, P., Sole, A., Mair, D., Cowton, T., King, M. A., & Palmer, S. (2011). Seasonal variations in greenland ice sheet motion: Inland extent and behaviour at higher elevations *Earth Planet. Sci. Lett.*, *307*(3-4), 271-278. doi: 10.1016/j.epsl.2011.04.014

Chandler, D. M., Wadham, J. L., Lis, G. P., Cowton, T., Sole, A., Bartholomew, I., ... Hubbard, A. (2013). Evolution of the subglacial drainage system beneath the Greenland Ice Sheet revealed by tracers *Nat. Geosci.*, *6*(3), 195-198. doi: 10.1038/ngeo1737

de Fleurian, B., Morlighem, M., Seroussi, H., Rignot, E., van den Broeke, M. R., Munneke, P. K., ... Tedstone, A. J. (2016). A modeling study of the effect of runoff variability on the effective pressure beneath Russell Glacier, West Greenland. *J. Geophys. Res*, *121*(10). doi: 10.1002/2016JF003842

Flowers, G. E. (2018). Hydrology and the future of the greenland ice sheet. *Nat. Comm.*, *9*(2729). doi: 10.1038/s41467-018-05002-0

Smith, L. C., Yang, K., Pitcher, L. H., Overstreet, B. T., Chu, V. W., Rennermalm, Å. K., ... Behar, A. E. (2017). Direct measurements of meltwater runoff on the greenland ice sheet surface. *Proceedings of the National Academy of Sciences*, *114*(50), E10622–E10631. doi: 10.1073/pnas.1707743114

---

## Referee Comment (RC2) · Anonymous Referee #2 · 19 Dec 2019

Review of: "Inter-comparison of surface meltwater routing models for the Greenland Ice Sheet and influence on subglacial effective pressures" by Yang et al.

General Comments: There has been a significant amount of recent work done on capturing processes influencing moulin hydrographs on sub-diurnal timescales. Going forward with these approaches will require significant investment of resources, particularly if field-derived empirical parameters are needed to calibrate supraglacial hydrology models. Underlying this work is an assumption that moulin discharge variability at sub-diurnal timescales might impact the evolution of inter- and sub-glacial hydrological networks, and thus ice dynamics. If this is the case, then supraglacial hydrological

processes necessitate further investment to be properly constrained at fine temporal and spatial scales. However, the impact of supraglacial discharge variability on subglacial hydrology has not yet been investigated, and it is therefore not clear if, where, and what specific investments are needed. In this context, this paper makes two major contributions:

1. This paper is a first attempt to investigate the extent to which moulin hydrographs matter for subglacial channel evolution and effective pressure on diurnal timescales. 2. This paper evaluates three different contemporary approaches to estimating daily moulin hydrographs and evaluates the consequences of each with respect to modelled evolution of subglacial channel evolution and effective pressure.

This paper therefore constitutes an original and valuable contribution to ongoing research on supraglacial hydrology. The paper could be improved by clarity and specificity around the methods used and the objectives of the paper. Suggestions in this respect are provided below and in the accompanying annotated PDF. My comments focus primarily on the supraglacial hydrology components of the study.

Specific comments:

Title: Only one of the models is a routing model. Consider saying 'inter-comparison of moulin hydrograph estimations' or something similar.

Throughout: The use of 'routing models' seems inaccurate. Only one (the SRLF) approach is a flow routing approach. The other two (RWF and SUH) do not route flow. A different word choice would be preferable. A 'comparison of hourly moulin discharge models', or something similar. . ..

Introduction: In general, I do not find that the introduction sets up the objectives of the paper very well. It does not provide sufficient information to set up a methods comparison, but also only emphasizes the subglacial channel evolution at the end – like an afterthought. I suggest the following changes:

[Figure]

- You need to be clearer in your introduction that this is not a methodological paper per se – as you say later in the paper, you cannot say which method performs better due to a lack of empirical evidence. You can only speculate on the modelled (not observed) hydrological implications of the three different methods. This needs to come across more strongly.

- The differences between the three approaches as well as the assumptions and limitations of each needs to be made clearer for a glaciology (rather than a hydrology) readership. The limitations of the empirically-derived RWF and SUH also need to be made clear, particularly the temporal limitations of the original field-derived moulin hydrograph measurement. As is, a comparison of these three approaches is only useful for conditions similar to those under which the SUH and RWF approaches are calibrated – this needs to be made clear.

- Because the comparison between the approaches is inherently limited due to the temporally-limited nature of the field moulin discharge measurements in the SUH and RWF, I think it would be useful in the introduction to put more emphasis on the goal of the paper as an exercise in examining (modelled) subglacial effective pressure sensitivity to diurnal hydrographs, rather than explicitly a comparison of moulin discharge estimate approaches. I would consider a bolder introductory statement around Page 2, Line 17 that frames the paper as a (preliminary) investigation of the extent to which moulin hydrograph estimates matter for affecting modelled subglacial hydrology and effective pressures.

- Last paragraph of introduction. I think there needs to be more explanation of what you are hoping to achieve with using the SHAKTI model. I think you need to be explicit that there is no objective away to compare the three moulin hydrograph methods, and the differences between them only matter in a glaciological sense if they significantly impact subglacial hydrology and effective pressure. You therefore run what is loosely a sensitivity test using SHAKTI to assess the modelled impacts of each approach. Although this comes out later in the paper, you need more framing of this consideration

in the introduction.

Study area and datasets:

- More specific justification of the chosen study IDCs is needed. They are approximately similar sizes to the IDC used in the Smith et al. (2017) measured moulin hydrographs, which should be pointed out, and they also appear to exclude large supraglacial lakes, which is likely to affect the comparison of the SRLF approach with the RWF and SUH approach. This should be noted in the study area description and in the discussion.

Methods: Overall, the methods seem written for hydrologists, not for glaciologists. More information is needed in this section to make it useful to its readers.

- Presumably, July 2015 was chosen for the MAR runoff simulations because that is coincident with the field-collection of the moulin discharge hydrograph. This should be made clear, so that the constraints on the method are obvious to readers.

- If this paper is to be a useful methodological resource for glacier hydrologists, a more complete comparison of the three approaches is needed. Perhaps a table would be useful in comparing the three methods – this table could keep track of the references, acronyms, assumptions, limitations, etc. . .

- Section 3.6 – your explanation of a 'dynamic' Ac is not clear, or perhaps I have missed it. In Figure 3, it looks like you tested all five of the different Ac values independently over the whole time span, but then there is also a 'dynamic Ac'. Is a 'dynamic' Ac one that evolves according to your six five-day RWF-UHs? Be sure to call that 'dynamic' here.

Results: the figures for this section are confusing. They need to be pulled apart to be more readable, and the legends and captions need better information.

- Figure 3 is too confusing with this many panels, and the result is that the lines are too small to make out the subtle differences due to the different variables. I suggest taking
the third column (effective pressure) out and putting it in its own figure, as it is a distinct part of the results and discussion. You could then make the main figure slightly larger, and show with a title on the legend in (g) and (j) that the different series refer to g) the DEM resolution and j) the channel initiation threshold (proxy for time).

- Figure 4 is not discussed in the results section. Some discussion should be provided in the 'long-term evolution' section, or it should be removed.

- Figure 3k – how is that there is so much smaller discharge for 5000m2 than there is for 100m2? With a higher Ac, there should be less efficient routing, lower peak Q and a flatter hydrograph but still, presumably, similar discharge. This distinction is not clear from the figure, perhaps because the lines are so compressed and the 'flashiness' of the hydrographs is not clear. Perhaps an inset figure would be helpful.

Discussion:

- Overall, I think this section is too critical of SRLF and not critical enough of SUH and RWF. Some discussion of the limitations of the latter two is needed, namely: the chosen study catchments do not appear to have lakes, and those methods may not perform adequately in catchments with lakes, at different times of year and in different snow/ice/surface slope conditions than those in which the field-measurements of the moulin hydrograph (Smith 2017) were collected.

- More discussion of the limitations of SHAKTI would be helpful. You should be clear that SHAKTI is used to provide preliminary insight into the possible importance of accurately capturing the details of an hourly moulin hydrograph, and that many complexities are not captured by this model.

Technical corrections: Please see specific in-text comments in the attached annotated PDF.

Please also note the supplement to this comment:

https://www.the-cryosphere-discuss.net/tc-2019-255/tc-2019-255-RC2-supplement.pdf

[Figure]

**Supplement:**

[revised manuscript text omitted]

---

## Author Comment (AC2) · 19 Jan 2020

**Reviewer #2**

General Comments: There has been a significant amount of recent work done on capturing processes influencing moulin hydrographs on sub-diurnal timescales. Going forward with these approaches will require significant investment of resources, particularly if field-derived empirical parameters are needed to calibrate supraglacial hydrology models. Underlying this work is an assumption that moulin discharge variability at sub-diurnal timescales might impact the evolution of inter- and sub-glacial hydrological networks, and thus ice dynamics. If this is the case, then supraglacial hydrological processes necessitate further investment to be properly constrained at fine temporal and spatial scales. However, the impact of supraglacial discharge variability on subglacial hydrology has not yet been investigated, and it is therefore not clear if, where, and what specific investments are needed. In this context, this paper makes two major contributions:

1) This paper is a first attempt to investigate the extent to which moulin hydrographs matter for subglacial channel evolution and effective pressure on diurnal timescales.

2) This paper evaluates three different contemporary approaches to estimating daily moulin hydrographs and evaluates the consequences of each with respect to modelled evolution of subglacial channel evolution and effective pressure.

This paper therefore constitutes an original and valuable contribution to ongoing research on supraglacial hydrology.

1. ("The paper could be improved by clarity and specificity around the methods used and the objectives of the paper. Suggestions in this respect are provided below and in the accompanying annotated PDF. My comments focus primarily on the supraglacial hydrology components of the study.")

**Reply:** We have better clarified the methods and the objective of the paper, as requested. We agree with the reviewer that the methods should "be made clearer for a glaciology (rather than a hydrology) readership" since the objective of the paper is to illustrate the impact of surface meltwater routing on subglacial effective pressure rather than to directly compare the three surface meltwater routing methods. We have carefully revised the supraglacial hydrology components of the study, as requested. We have also included Table S1, which simply explains the benefits of using different routing models.

2. ("Specific comments: Title: Only one of the models is a routing model. Consider saying 'inter-comparison of moulin hydrograph estimations' or something similar. Throughout: The use of 'routing models' seems inaccurate. Only one (the SRLF) approach is a flow routing approach. The other two (RWF and SUH) do not route flow. A different word choice would be preferable. A 'comparison of hourly moulin discharge models', or something similar....")

**Reply:** We suggest that the three models (SUH, RWF, and SRLF) are all routing models. A routing model does not need to explicitly determine how meltwater produced on a cell is routed to its downstream cell(s) in a catchment, which is the aim of spatially-distributed routing models. In contrast, a routing model can be lumped and only determines how surface meltwater produced in the catchment is temporally routed to the catchment outlet.

Unit hydrograph (UH) is designed for this purpose. UH is a transfer function that models catchment runoff response to rainfall (melt in our case) events for some unit duration and unit depth of effective water input. In this study, all the three models (SUH, RWF, and SRLF) exhibit their UHs and consequently can temporally route surface meltwater to the catchment outlet and yield moulin discharge hydrographs. From this perspective, we suggest that they are all routing models.

Moreover, RWF and SRLF actually work very similarly. SRLF uses Manning's open-channel equation to calculate meltwater flow velocity for each cell of a catchment, while RWF uses constant hillslope and open-channel flow velocities calibrated from field measurements to determine each cell's flow velocity. As such, RWF and SRLF both generate a velocity raster. Then, integrating velocity with flow path distance calculated from DEM, meltwater transport time from each cell to the catchment outlet can be determined and the transport time distribution yields UH. In short, the primary difference between RWF and SRLF is the way they determine flow velocity for each cell.

We have added Table S1 to summarize the benefits of using different routing models and better illustrated the three routing models in the methods and data section.

[revised manuscript text omitted]

3. ("Introduction: In general, I do not find that the introduction sets up the objectives of the paper very well. It does not provide sufficient information to set up a methods comparison,

but also only emphasizes the subglacial channel evolution at the end – like an afterthought. I suggest the following changes: - You need to be clearer in your introduction that this is not a methodological paper per se – as you say later in the paper, you cannot say which method performs better due to a lack of empirical evidence. You can only speculate on the modelled (not observed) hydrological implications of the three different methods. This needs to come across more strongly.")

**Reply:** We have included the following statement in the Introduction: "Notably, this study cannot, in good faith, focus on method comparison, nor determine the 'best' (i.e., best able to reproduce a real-world moulin hydrograph) due to the lack of calibration and validation data. Owing to this limitation, the goal of this study is to assess differences among the three meltwater routing models, rather than revealing which model most realistically simulates surface meltwater routing on the ice surface. By using the outputs from all three as meltwater inputs to drive the SHAKTI subglacial model, we characterize the impact of their differences on subglacial effective pressure, and, more generally, the importance of routing supraglacial runoff on subglacial conditions." Moreover, we have further emphasized this fact in the discussion and is generally discussed in section 4.5 (Future research directions).

4. ("The differences between the three approaches as well as the assumptions and limitations of each needs to be made clearer for a glaciology (rather than a hydrology) readership. The limitations of the empirically-derived RWF and SUH also need to be made clear, particularly the temporal limitations of the original field-derived moulin hydrograph measurement. As is, a comparison of these three approaches is only useful for conditions similar to those under which the SUH and RWF approaches are calibrated – this needs to be made clear.")

**Reply:** We have better explained the three approaches and made them clearer for the glaciology readership, including the addition of Table S1. The limitations of the empirically-derived RWF and SUH have been made clearer, as requested.

SUH and RWF both rely on several empirically parameters ( $C_p$  and  $C_t$  for SUH, and  $v_h$  and  $v_c$  for RWF) calibrated from a moulin hydrograph measured at the Rio Behar catchment, southwest GrIS during a very short time period (72 hours), July 2015 (Smith et al., 2017). In contrast, SRLF is more solid because it only relies on DEM to calculate meltwater flow velocities (Banwell et al., 2012). In this study, we assume these empirically parameters are transferable over space and time but this assumption needs further validation. It may hold for ice sheet surface with similar hydrologic and glaciological environments but it may be problematic to apply over larger space and longer time. A second independent, long-term moulin hydrograph will help to address this problem.

5. ("Because the comparison between the approaches is inherently limited due to the temporally-limited nature of the field moulin discharge measurements in the SUH and RWF, I think it would be useful in the introduction to put more emphasis on the goal of the paper as an exercise in examining (modelled) subglacial effective pressure sensitivity to diurnal hydrographs, rather than explicitly a comparison of moulin discharge estimate approaches. I

would consider a bolder introductory statement around Page 2, Line 17 that frames the paper as a (preliminary) investigation of the extent to which moulin hydrograph estimates matter for affecting modelled subglacial hydrology and effective pressures.")

**Reply:** We have included two modifications to the introduction to emphasize the importance of the supraglacial system on subglacial hydrology. First, the current lack of representation of the surface meltwater routing leads to an insufficient understanding of surface-to-bed meltwater connections and ice dynamics, particularly on diurnal timescales. Therefore, constraints on IDC discharge can provide critical boundary conditions for studies of the subglacial hydrologic system. Second, by using the outputs from all three as meltwater inputs to drive the SHAKTI subglacial model, we have characterized the impact of their differences on subglacial effective pressure, and, more generally, the importance of routing supraglacial runoff on subglacial conditions. We have also emphasized the subglacial results in the last line of the abstract. We thoroughly discuss the implications of these different surface meltwater routing models on subglacial hydrology in section 4.2.

6. ("Last paragraph of introduction. I think there needs to be more explanation of what you are hoping to achieve with using the SHAKTI model. I think you need to be explicit that there is no objective away to compare the three moulin hydrograph methods, and the differences between them only matter in a glaciological sense if they significantly impact subglacial hydrology and effective pressure. You therefore run what is loosely a sensitivity test using SHAKTI to assess the modelled impacts of each approach. Although this comes out later in the paper, you need more framing of this consideration in the introduction.")

**Reply:** See our reply to your comment 3. We have revised the last paragraph of introduction to better illustrate the objective of this study, as requested and explicitly indicate our goals with the lines: "By using the outputs from all three routing models as meltwater inputs to drive the SHAKTI subglacial model, we seek to characterize the impact of differences in surface routing on subglacial pressures and evolution, particularly over diurnal timescales. More generally, these results can demonstrate the extent to which the choice of surface meltwater routing algorithms can alter modelled subglacial conditions."

7. ("Study area and datasets: More specific justification of the chosen study IDCs is needed. They are approximately similar sizes to the IDC used in the Smith et al. (2017) measured moulin hydrographs, which should be pointed out, and they also appear to exclude large supraglacial lakes, which is likely to affect the comparison of the SRLF approach with the RWF and SUH approach. This should be noted in the study area description and in the discussion.")

**Reply:** We have better explained the reasons to select the four IDCs, as requested: "They are distributed at approximately 200 m elevation intervals in order to span the elevational range of most well-developed IDCs found in the Russell Glacier region and the variable surface melt conditions of this region (Yang and Smith, 2016). Large supraglacial lakes are absent in these four IDCs (Figure 1) and surface meltwater is all routed to the moulin at the catchment outlet. As such, surface runoff produced in each IDC equals to the moulin discharge (Smith et al.,

2017). A moulin discharge hydrograph collected at Rio Behar catchment (IDC2 in our study), southwestern GrIS (67.049346N, 49.025809W) for 72 h from 20 to 23 July 2015 was used to calibrate key parameters of SUH and RWF models. It is problematic to apply these empirically-derived parameters over large spaces and long times (Yang et al., 2018). Therefore, the four IDCs distributed in a relatively small region were selected and the areas of IDC1, IDC3, and IDC4 are similar to the Rio Behar catchment (IDC2)."

We have better illustrated the limitations of SUH and RWF in the discussion section, as requested: "SUH and RWF both rely on several empirically parameters ( $C_p$  and  $C_t$  for SUH, and  $v_h$  and  $v_c$  for RWF) calibrated from a moulin hydrograph measured at the Rio Behar catchment, southwest GrIS during a very short time period (72 hours), July 2015 (Smith et al., 2017). In contrast, SRLF is more solid and applicable over large spaces and long times because it only relies on DEM to calculate meltwater flow velocities (Banwell et al., 2012). In this study, we assume these empirically parameters are transferable over space and time but this assumption needs further validation. It may hold for ice sheet surface with similar hydrologic and glaciological environments but is problematic to apply over larger space and longer time. A second independent, long-term moulin hydrograph will help to address this problem."

**8. ("Methods: Overall, the methods seem written for hydrologists, not for glaciologists. More information is needed in this section to make it useful to its readers.")**

**Reply:** We have better introduced the three routing models from a perspective of glaciologists. A new section has been added to explain Unit Hydrograph and its application for calculating moulin discharge. Additional new text has been added to better explain Snyder Synthetic Unit Hydrograph, Surface Routing and Lake Filling, and Rescaled Width Function, as requested.

9. ("Presumably, July 2015 was chosen for the MAR runoff simulations because that is coincident with the field-collection of the moulin discharge hydrograph. This should be made clear, so that the constraints on the method are obvious to readers.")

**Reply:** Yes, July 2015 runoffs were derived to be coincident with the field-collection of the moulin discharge hydrograph. Additional new text has been added to explain this point, as requested.

10. ("If this paper is to be a useful methodological resource for glacier hydrologists, a more complete comparison of the three approaches is needed. Perhaps a table would be useful in comparing the three methods – this table could keep track of the references, acronyms, assumptions, limitations, etc...")

**Reply:** Thanks for this great comment. We have added a new table (Table S1) to better compare the three routing models, as suggested (see our reply to your comment 2).

11. ("Section 3.6 – your explanation of a 'dynamic' Ac is not clear, or perhaps I have missed it. In Figure 3, it looks like you tested all five of the different Ac values independently over the whole time span, but then there is also a 'dynamic Ac'. Is a 'dynamic' Ac one that evolves according to your six five-day RWF-UHs? Be sure to call that 'dynamic' here.")

**Reply:** Yes, we tested all five of the different  $A_c$  values independently over the whole time span and then a 'dynamic'  $A_c$  one evolves according to the six five-day RWF-UHs, as the reviewer pointed out. We have added a new sentence "Each RWF UH was conducted to calculate moulin discharge for five days and the resultant moulin discharge is termed as dynamic  $A_c$  discharge" to better explain dynamic  $A_c$ , as requested.

**12. ("Results: the figures for this section are confusing. They need to be pulled apart to be more readable, and the legends and captions need better information.")**

**Reply:** We have carefully revised the figures to make them more readable, as requested. See our reply to your following comments.

13. ("Figure 3 is too confusing with this many panels, and the result is that the lines are too small to make out the subtle differences due to the different variables. I suggest taking the third column (effective pressure) out and putting it in its own figure, as it is a distinct part of the results and discussion. You could then make the main figure slightly larger, and show with a title on the legend in (g) and (j) that the different series refer to g) the DEM resolution and j) the channel initiation threshold (proxy for time).")

**Reply:** We have taken the third column (effective pressure) and put it in its own figure, as the reviewer suggested (Figures S1 and S2). We have made Figure 3 larger and reshaped the width ratio between the first column (UH) and the second column (moulin discharge) from 1:1 to 1:1.5 to better represent the diurnal moulin discharge. We have added a title "DEM resolution" to Figure 3c and 3e, and a title "Ac" to Figure 3g, changed the x-axis labels of Figure 3e and 3g into "RWF UH ( $A_c = 100 \text{ m}^2$ ) and "RWF UH (2 m DEM), respectively, and added legend "RCM runoff" to Figure 3b, 3d, and 3f. We have better explained "the channel initiation threshold (proxy for time)" in the caption of Figure 3.

---

## Author Response (AR1)

March 5, 2020

Dear Editor Elizabeth Bagshaw,

Thank you for your letter on February 13 inviting revisions to our manuscript, *"Inter-comparison of surface meltwater routing models for the Greenland Ice Sheet and influence on subglacial effective pressures."* We are pleased to state that we have complied with all of the requests made by you and the two reviewers. A stepwise, detailed response to all comments is as follows:

**Handling Editor:**

Thank you for your comments to the reviewers. They note that the paper is a valuable contribution to the field, and I encourage you to take into account their recommendations in your preparation of the revised article.

In particular, please address:

1. Overuse of acronyms, particularly in the abstract and subheadings

2. Readability of the figures. I note that even in the revised versions some of the titles are superimposed.

3. Framing of the study for a glaciology audience

4. Boundary conditions.

**Reply:** Thanks for your comment. We have: (1) deleted most acronyms in the abstract and subheadings (p. 1); (2) revised the superimposed titles in Figures 2, 4, and S1-S4 (p. 24-26); (3) better explained the three surface meltwater routing methods, making the study more useful and readable for a glaciology audience (p. 4, lines 2-15; p. 5, lines 13-31; p. 6, lines 1-16; p. 7, lines 1-13; p. 8, lines 2-8); and (4) rerun the subglacial hydrology simulations with a more realistic downstream boundary condition (p. 9, lines 5-16; p. 10, lines 16-32; p. 11, lines 1-32), as requested.

**Reviewer #1:**

General comments: This study focuses on different ways of treating the supraglacial drainage of water at the surface of ice sheets. The region of interest studied here is Russell Glacier in South West Greenland. The authors presents an inter-comparison of three different surface routing models and compare their results to the output of a Regional Climate Model (RCM). The different inputs are further compared through by using them as the forcing quantity provided to a subglacial drainage model. The conclusion of the study are that the use of a supraglacial drainage system allows to get a better representations of the lag of the water input to the moulins. Some sensitivity among models also allow to quantify the impact of the

Digital Elevation Model resolution on the drainage characteristics.

1. ("This study provides an interesting insight into the differences that arise from the use of different supraglacial drainage model. However, my impression is that this study should be further refined in order to be more understandable and provide a usable tool for the community. I find that the presentation of the different models and their results is lacking detail and clarity. Moreover I am concerned by the choice that were made with regards to the boundary condition that are applied to the subglacial hydrology model. Details of my concerns and potential improvement are given bellow section by section.")

**Reply:** Thanks for the comment. We have: (1) reorganized the results and the discussion sections to make the study more understandable; (2) better clarified the three surface meltwater routing models in the methods and data section (p. 4, lines 2-15; p. 5, lines 13-31; p. 6, lines 1-16; p. 7, lines 1-13; p. 8, lines 2-8); (3) rerun the subglacial hydrology simulations with a more realistic downstream boundary condition (p. 9, lines 5-16; p. 10, lines 16-32; p. 11, lines 1-32); (4) revised the main figures of this study to make them more readable (p. 24-28); and (5) published all the model codes online (https://doi.org/10.6084/m9.figshare. 11635932.v1). We think the revised manuscript is much easier to follow and the three routing models will be usable tools for the community.

2. ("1.1 Abstract: The abstract is quite hard to read due to the accumulation of acronyms. Where possible I would urge the authors to refrain from using acronyms in this part of the papers. It might be beneficial to simplify the abstract to make it more accessible to readers which might afterwards gather the details of the study in the rest of the paper. As an example, the author could state that they compare three surface meltwater routing models at this point 1 without specifying those models, the list of variables line 21 (page 1) could be omitted and replaced by "key variables".)

**Reply:** We have simplified the abstract, as requested (p. 1, lines 14-33; p. 2, lines 1-6). Most acronyms have been deleted. We suggest that the full names of the three routing models are necessary because they are used to explain the results. The revised abstract is as follows:

"Each summer, large volumes of surface meltwater flow over the Greenland Ice Sheet surface and drain through moulins to the ice sheet bed, and impact subglacial hydrology and ice flow dynamics. Runoff modulations, or routing delays due to ice sheet surface conditions, propagate to englacial and subglacial hydrologic systems and require accurate assessment to correctly estimate subglacial effective pressures and short-term lags between surface meltwater production and ice motion. This study compares hourly supraglacial moulin discharge simulations from three surface meltwater routing models, (1) synthetic unit hydrograph, (2) surface routing and lake filling, and (3) rescaled width function, for four internally drained catchments located on the southwestern Greenland ice sheet surface. Using surface runoff from the Modèle Atmosphérique Régionale regional climate model (RCM), simulated variables used for surface meltwater routing are compared among the three routing models. For each catchment, simulated moulin hydrographs are used as input to the SHAKTI subglacial hydrologic model to produce corresponding subglacial effective pressure variations in the vicinity of a single moulin. Two routing models, surface routing and lake filling and rescaled width function, which require the use of a digital elevation model

(DEM), are assessed for the impact of DEM spatial resolution on simulated moulin hydrographs. Surface routing and lake filling is sensitive to DEM spatial resolution, whereas rescaled width function is not. Our results indicate the three surface meltwater routing models perform differently in simulating moulin peak discharge and time to peak, with rescaled width function simulating later, smaller peak moulin discharges than synthetic unit hydrograph or surface routing and lake filling. We also demonstrate that the seasonal evolution of supraglacial stream/river networks can be readily accommodated by rescaled width function but not synthetic unit hydrograph or surface routing and lake filling models. Overall, all three models produce more realistic supraglacial discharges than simply using RCM runoff outputs without an applied routing scheme; however, there are significant differences in supraglacial discharge generated by the three models tested. This variability among surface meltwater routing models is reflected in SHAKTI subglacial hydrology simulations, which yield substantially different diurnal effective pressure amplitudes depending on the applied surface meltwater routing model; however, display relatively consistent mean effective pressure across routing models."

3. ("1.2 Introduction: The introduction gives a succinct outlook on the motivations of the study. This could be developed further to point out the current lack of representation of the supraglacial drainage system and the necessity to have a better representation of this system. The description starting on line 20 (page 2) would be better in a method section of the paper. Moreover, some terms defined in the introduction (such as Unit Hydrograph or Internally Drained Catchments) might not be familiar to the Cryosphere community and the author should consider defining those in more details.")

**Reply:** The current lack of representation of the surface meltwater routing leads to an insufficient understanding of surface-to-bed meltwater connections and ice dynamics, as the reviewer pointed out. Additional new text has been added to better highlight this point, as requested (p. 2, lines 28-30).

The definitions of Unit Hydrograph and Internally Drained Catchments have been added, as requested (p. 2, lines 19-20; p. 3, lines 3-4). Internally drained catchments (IDCs) are "hydrologic units on the GrIS surface that collect and drain meltwater through supraglacial stream/river networks to terminal moulins or lakes" (Yang and Smith, 2016). Unit hydrograph (UH) is "a transfer function that is widely used for modeling catchment runoff response to rainfall events for some unit duration and unit depth of effective water input" (Smith et al., 2017). A new section has been added to better define unit hydrograph in more details, as requested (p. 5, lines 13-23).

The description starting on line 20 (page 2) introduces the three routing models and their different assumptions and data requirements. We suggest that it may be useful to explain these three models clearly in the introduction section.

4. ("I don't completely agree with the statement starting on line 10 (page 2) to my knowledge supraglacial meltwater routing is usually simplified in subglacial hydrology models (e.g. Banwell et al., 2016; de Fleurian et al., 2016) it would be interesting to have

some citation here that present studies directly using an RCM as their water input. I am not sure that the citation to Flowers et al. (2018) is relevant in this context or I missed the point of the author here. Further down, the citation to Bartholomew et al. (2011) seemed to be misplaced here as this specific study treats about observations rather than modelling.")

**Reply:** Flowers (2018) is a review of Greenland hydrology, which discusses some important issues in surface-to-bed meltwater connection. We agree with the reviewer that Flowers (2018) may not be appropriate to state that "surface meltwater routing is either simplified or simply ignored" so it has been deleted, as requested (p. 2, line 18). Reviewer 2 made the same comment. We agree that Bartholomew et al. (2011) is primarily "about observations" so have removed this citation (p. 2, line 21). We have listed several studies (see Table S1) that use RCM runoff as water input, as requested.

5. ("1.3 Study area and data source: This section is missing a major information as the study area is actually never named. The Russell glacier region will be familiar to most of the reader interested of the subject but mention of it should still appear in the paper. My opinion is that this section should be merged into a section 3 (Methods and Data). Regarding the content of the present section it is not clear to me how and why the IDCs that are presented in this section were generated and why those specific IDCs have been chosen.")

**Reply:** Section 2 (Study area and data sources) has been merged into Section 3 (Methods and Data) as suggested (p. 4, line 19). The Russell Glacier region has been added to better introduce the study area, as requested (p. 4, lines 20-24).

We have better explained the reasons to select the four IDCs, as requested: "They are distributed at approximately 200 m elevation intervals in order to span the elevational range of most well-developed IDCs found in the Russell Glacier region and the variable surface melt conditions of this region. Large supraglacial lakes are absent in these four IDCs and surface meltwater is all routed to the moulin at the catchment outlet. As such, surface runoff produced in each IDC should equal to the moulin discharge. A moulin discharge hydrograph collected at Rio Behar catchment (IDC2 in this study; 67.049346N, 49.025809W) for 72 h from 20 to 23 July 2015 was used to calibrate parameters of SUH and RWF models (see Sections 2.5 and 2.6). It is problematic to apply these empirically-derived parameters over large spaces and long times (Yang et al., 2018). Therefore, the selected IDCs are distributed in a relatively small region and the areas of IDC1, IDC3, and IDC4 are similar to the Rio Behar catchment (IDC2)", as requested (p. 4, lines 23-31).

6. ("1.4 Methods: The description of the different models here is quite brief and some more details could be provided. Particularly it would be interesting to have a better overview of the advantages and drawbacks of each models. The paragraph starting line 15 (page 5) would fit better in the introduction of the study rather than here. Subsections 3.5 and 3.6 refer to the sensitivity studies that where performed for some of the model, it could be beneficial to transfer those sections into the descriptions of the relevant models. That would outline the advantages and potential drawbacks of the models and would clarify the overall setup of the experiments.")

**Reply:** Overview of the advantages and drawbacks of each model has been provided in the introduction section and a new table (Table S1) has been added to better compare the three routing models. RWF can mimic seasonal evolution of supraglacial stream/river networks by varying the partitioning of hillslope versus open-channel zones. SRLF, in contrast, assumes the bare ice surface has a stable response to the surface melt and uses static meltwater routing velocities to build the UH. SUH relies on $C_p$ and $C_t$ to build the UH. If we can have a moulin hydrograph during the entire melt season, we can calculate multi-temporal $C_p$ and $C_t$ using variable time-to-peak ($t_p$), peak discharge ($h_p$), and main-stem stream length ($L$), thus creating multi-temporal UHs. Unfortunately, we do not have such measurements at present so SUH cannot mimic variable hydrologic response of a catchment to surface melt.

**Table S1.** A brief summarization of surface meltwater routing models.

| Model | Meltwater Routing | Applicable on bare ice surfaces | | Applicable on snow surfaces | Parameter dependency | DEM dependency | Case study |
|---|---|---|---|---|---|---|---|
| | | Hillslope | Open-Channel | | | | |
| **Instantaneous RCM runoff** | No | - | - | Yes | No | No | (McGrath et al., 2011; Bartholomew et al., 2012; Rennermalm et al., 2013; Fitzpatrick et al., 2014) |
| **Snyder Synthetic Unit Hydrograph (SUH)** | Yes | - | - | Yes, but model parameters should be recalibrated | $C_p$, $C_t$ are calibrated using a field-measured moulin hydrograph | No | (Smith et al., 2017) |
| **Surface Routing and Lake Filling (SRLF)** | Yes | No | Yes | Yes | No | DEM is required to calculate meltwater flow velocities for all catchment cells | (Arnold et al., 1998; Willis et al., 2002; Banwell et al., 2012; Banwell et al., 2013; Arnold et al., 2014; Banwell et al., 2016; de Fleurian et al., 2016; Koziol and Arnold, 2018) |
| **Rescaled Width Function (RWF)** | Yes | Yes | Yes | Yes, but model parameters should be recalibrated | Hillslope and open-channel flow velocities ($v_h$ and $v_c$) are calibrated using a field-measured moulin hydrograph | High-resolution (<10 m) DEM is required to calculate hillslope flow path length | (Yang et al., 2018) |

Moreover, SUH and RWF both rely on several empirically parameters ($C_p$ and $C_t$ for SUH, and $v_h$ and $v_c$ for RWF) calibrated from a moulin hydrograph measured at the Rio Behar catchment, southwest GrIS during a very short time period (72 hours), July 2015 (Smith et al., 2017). In contrast, SRLF is more solid and applicable over large spaces and long times. because it only relies on DEM to calculate meltwater flow velocities (Banwell et al., 2012). In this study, we assume these empirically parameters are transferable over space and time but this assumption needs further validation. It may hold for ice sheet surface with similar hydrologic and glaciological environments but is problematic to apply over large regions and long times due to evolving ice surface characteristics. Multiple independent, long-term

moulin hydrographs will help eliminate the need for this assumption. We have further illustrated the advantages and drawbacks of these routing models through the manuscript (p. 6, lines 3-6; p. 7, lines 9-12; p. 8, lines 2-8; p. 14, lines 6-13).

The paragraph starting line 15 (page 5) has been removed to the introduction section, as requested (p. 4, lines 6-15).

Subsections 3.5 and 3.6 are not sensitivity studies as the reviewer suggested. In contrast, they are both important topics in terrestrial hydrology and this study attempts to expand these two topics to ice sheet hydrology. Zhang and Montgomery (1994) is a classic study investigating the impact of DEM spatial resolution on terrestrial stream flow and we followed their method to investigate the impact of DEM spatial resolution on surface meltwater routing. The temporal evolution of stream/river networks on stream flow is a state-of-the-art topic and attracts growing attention in terrestrial hydrology. For example, a recent study shows that the extension and retraction of the terrestrial stream network can substantially change the mean travel time and the shape of the travel time distribution (van Meerveld et al., 2019), similar to the finding of our study. Therefore, we suggest that these analyses are partially independent of routing models and should have their own subsections. We have better explained the importance of these two sections (p. 12, lines 15-19; p. 13, lines 7-9; p. 16, lines 9-12).

7. ("From the references that were provided in the paper regarding the SRLF model I understand that this model is routing water with different equations if it sits on snow or on bare ice. From the model description given here it seems that only the bare ice formulation was used. Is that so? If yes the reasons for this choice should be explained.")

**Reply:** Yes, only the bare ice formulation of the SRLF model was used. The parameters of SUH and RWF routing models were calibrated using field-measured moulin discharge on bare ice (Smith et al., 2017; Yang et al., 2018). Therefore, to make these meltwater routing models comparable, we only discussed the situation of meltwater routing on the bare ice surface. This point has been better explained in the revised manuscript, as requested (p. 6, lines 10-16).

8. ("As stated above my main concern with this study is the way in which the subglacial hydrology model is set-up. In my opinion the boundary condition that is given for the left edge of the domain is not realistic, I do not think that we expect to find water pressure at the atmospheric pressure anywhere under the ice sheet. A more sensible choice would be to set the water pressure at a given fraction of the overburden pressure. A change of boundary condition would need to perform new simulations but I would expect a good argumentation on the choice of the present boundary condition if it is to be kept. I also do not understand why the slopes of the bed and surface, and velocities are not taken from the values of the IDCs as is done for the ice thickness. As it stands now I have a hard time trusting the results from the subglacial hydrology model as it seems that the downstream boundary that is currently set is exerting an important control on the whole domain. I would also note that the Figure 2 related to this section is not very informative and could probably be omitted.")

**Reply:** Thank you for the recommendation to use a more realistic boundary condition for the moulin input-forced subglacial hydrology simulations in the interior of the ice sheet. We have rerun all SHAKTI simulations using a downstream boundary condition with water pressure corresponding to 50% of ice overburden pressure. The new simulations also use mean surface slope calculated for each catchment drainage (used for both surface and bed slopes, maintaining a uniform slab of ice for the 1 km square domain), as well as sliding velocity corresponding to 100% of the mean annual observed surface velocity in each drainage catchment. The mean slopes and surface velocities are included in Table S2. Figure 2 was originally included to show the model discretization and clearly indicates the moulin location at the bed, but we have removed it, as suggested. We have revised "Section 2.9 Subglacial hydrology modelling" and "3.2 Simulations of subglacial effective pressure" to better explain the methods and results of subglacial hydrology simulations (p. 8, lines 22-33; p. 9, lines 1-16; p. 10, lines 16-32; p. 11, lines 1-32).

**Table S2.** Summary of four study catchments.

| Catchment ID | | IDC1 | IDC2 | IDC3 | IDC4 |
|---|---|---|---|---|---|
| Area (km²) | | 53.0 | 66.9 | 57.3 | 58.5 |
| Mean Elevation (m) | | 1054 | 1248 | 1473 | 1646 |
| Mean surface slope (m/m) | | 0.018 | 0.020 | 0.008 | 0.008 |
| Mean bed elevation (m) | | 207 | 309 | 247 | 222 |
| Mean bed slope (m/m) | | 0.050 | 0.075 | 0.036 | 0.022 |
| Mean ice thickness (m) | | 847 | 939 | 1226 | 1424 |
| Mean ice flow velocity (m/a) | | 116 | 99 | 98 | 73 |
| Distance to ice edge (km) | | 25 | 40 | 70 | 100 |
| Peak discharge time | RCM | 13-15 | 13-15 | 13-14 | 13-14 |
| | SUH | 19-20 | 20-21 | 19-20 | 19-20 |
| | SRLF 2m, 5m, 10m, 30m, 90m | 18-19, 20-21, 21-22, 22-23, 22-23 | 18-19, 19-20, 21-22, 21-22, 21-22 | 17-18, 18-19, 20-21, 22-23, 22-23 | 17-18, 19-20, 23, 23, 23 |
| | RWF | 22-23 | 21-22 | 20-21 | 21-22 |

9. ("1.5 Results: In general I find the presentation of the results quite hard to follow. This might come from the structure that was chosen by the author, from the presentation of the figures or both. I also wonder why only the results from IDC 1 are presented, it appears from the supplementary figures that the results from the four IDCs are quite similar but this should be stated. I also expect that changing the boundary conditions and parameters of the subglacial hydrology model may alter those results. Regarding the presentation of the results, it would be clearer to me if the author would describe first the results of the inter-comparison itself before delving into the sensitivity studies that were performed on DEM resolution and the value of Ac. Comparing the results to a given reference might also help with the clarity of the text.")

**Reply:** The structure of the results section has been reorganized, as requested. The revised results section includes: 4.1 Simulations of supraglacial moulin discharge, 4.2 Simulations of subglacial effective pressure, 4.3 Long-term evolution of moulin discharge simulations, and 4.4 Effects of DEM spatial resolution on surface meltwater routing. The sub-sections of discussion have been changed accordingly. We agree with the reviewer that the new structure is much easier to follow. The results from the four IDCs are quite similar as the reviewer pointed out. We have explicitly stated this point: "The three UHs (i.e., SUH, SRLF UH,

and RWF UH) and their routed moulin discharges for the four IDCs are presented in the supplement; the resultant patterns of moulin hydrographs are similar so we use IDC1 to illustrate our results here", as requested (p. 10, lines 2-5).

As you might expect, the modifications to the subglacial hydrology simulations to include the more realistic boundary condition, surface slopes, and sliding velocities do change the magnitude of the resulting effective pressures (now lower than with the atmospheric pressure boundary condition used in the initial submission). The text has been updated to describe the new results, and the overall behavior and findings of the study remain consistent (p. 8, lines 22-33; p. 9, lines 1-16; p. 10, lines 16-32; p. 11, lines 1-32).

10. ("I generally find the presented figures a bit too busy and so hard to read. Figure 3 is described a lot throughout the manuscript but the size of some panels make it hard to read. As for the text having specific figures for the inter-comparison and the sensitivity study might help to lighten the figures. Figure 4, 6 and 7 however are not described in the Results part and should be included there.")

**Reply:** To make Figure 3 more readable, we have taken the third column (subglacial effective pressure) and put it in its own figure, as it is a distinct part of the results and discussion (Figures S1 and S2). We have made Figure 3 larger and reshaped the width ratio between the first column (UH) and the second column (moulin discharge) from 1:1 to 1:1.5 to better represent the diurnal moulin discharge (Figure S1). We have added a title "DEM resolution" to Figure 3c and 3e, and a title "$A_c$" to Figure 3g, changed the x-axis labels of Figure 3e and 3g into "RWF UH ($A_c$ = 100 m$^2$)" and "RWF UH (2 m DEM), respectively, and added legend "RCM runoff" to Figure 3b, 3d, and 3f to make the figure easy to follow. Figure 4 has been removed as suggested. Figures 6 and 7 have been updated and better explained in the main text, as requested (Figures S3 and S4).

Moreover, we cannot obtain the "optimal simulations for SRLF and RWF" because determining DEM spatial resolutions or cumulative area thresholds are important topics in surface hydrology rather than sensitivity analysis (see our reply to your comment 6). Therefore, we prefer to plot variable simulations together and believe necessary legends make all the sub-figures understandable. However, we agree with the reviewer that a less busy figure will be easier to follow. Therefore, we only plot simulation results using 2 m DEM in Figure S4.

[Figure]

**Figure S1.** Presentation of Unit Hydrographs (UHs) (column 1) and moulin discharges (column 2) of IDC1 during July 2015, as simulated by three supraglacial routing models (SUH, SRLF, and RWF). $A_c$ is the cumulative contributing area required to initiate a supraglacial meltwater channel and dynamic $A_c$ values are used as proxy for time to simulate the temporal evolution of supraglacial stream/river networks. Simultaneous RCM runoff (grey line) is shown to indicate the effect of surface meltwater routing process on moulin discharge.

[Figure]

**Figure S2.** Effective pressures for IDC1 simulated by SHAKTI, with inputs to the subglacial system via a single moulin prescribed by the moulin discharges (shown in Figure S1) calculated by the various routing models. The effective pressure shown here is the spatial mean for the entire 1-km square domain which contains the moulin input at its center.

[Figure]

**Figure S3.** Snapshots of subglacial hydrology fields on day 23 in IDC1 using the SUH routing method to drive moulin input (see full animation of channel evolution and fluctuation in the supplement). An efficient channelized drainage pathway develops from the moulin location at the center of the domain to the outflow at the left, characterized by higher gap height, water flux, effective pressure, and lower hydraulic head than its surroundings perpendicular to flow.

[Figure]

**Figure S4.** The average two-day cycle of moulin discharge (Q) for IDC1 during July 2015. The daily minimum input in supraglacial moulin discharge (solid lines) corresponds generally to maximum effective pressure (dashed lines), and is followed within 8-9 hours by the daily minimum effective pressure (maximum subglacial water pressure). This suggests that the system shuts down due to creep with low meltwater input, and becomes highly pressurized as meltwater input increases again. As the new water inputs are accommodated, efficient pathways reform and effective pressure increases (subglacial water pressure decreases).

11. ("Lastly I have not seen any information with regard to the sampling of the effective pressure that is discussed, is it an average value or this value is taken at a specific point?")

**Reply:** The effective pressure that was described in the initial submission corresponded to the value at the moulin location on the bed (i.e. the head of the channel that forms to the downstream boundary). This description may have been inadvertently removed in the initial manuscript submission (our apologies). In the revised manuscript, we have altered the figures and discussion to focus instead on the mean effective pressure for the entire domain. While the spatial mean effective pressure variations are not as dramatic as seen at the moulin location itself, we feel this gives a more informative view of effective pressure behavior in the vicinity of a moulin, which is potentially more useful from an application perspective.

12. ("1.6 Discussion and Conclusion: The discussion of the manuscript is clear, it would however take advantage of the alterations suggested above for the Result section. Particularly describing all the figure in more details in the result section would help during the discussion. I also expect that the changes required above regarding the subglacial hydrology model might have a significant impact on the results and should be taken into account in the discussion.")

**Reply:** The discussion section has been reorganized, as requested. The revised discussion section includes: 5.1 Implications of surface meltwater routing method inter-comparison, 5.2 Influence on diurnal subglacial pressure variations, 5.3 Influence of seasonal supraglacial drainage evolution on meltwater routing, 5.4 Impact of DEM resolution on supraglacial meltwater routing, and 5.5 Future research directions of surface-to-bed meltwater connection. We have better described all the figures in the results section, as requested (see our reply to your comment 10).

As described above, the subglacial hydrology simulations have been rerun with more appropriate/realistic boundary conditions, slopes, and sliding velocities. As noted, these changes do influence the magnitude of resulting effective pressures, but not the overall behavior and differences between routing methods. The text has been updated to reflect the new results (p. 10, lines 16-32; p. 11, lines 1-32).

13. ("I have noted a few minor concern on this section which are listed in the Specific comments bellow. Specific comments. Below is a list of more specific comments throughout the manuscript given with line and page number: Page 1, Line 16: "ice surface" can be replaced by "ice sheet surface".")

**Reply:** Changed as requested (p. 1, line 16).

14. ("Line 17: "climatological melt" should be replaced by "surface meltwater".")

**Reply:** Changed as requested (p. 1, lines 17-18).

15. ("Line 21: MAR abbreviation is not defined here.")

**Reply:** MAR has been replaced by "Modèle Atmosphérique Régionale" (p. 1, line 21).

16. ("Line 23: "input" can be replaced by "used as input"")

**Reply:** Changed as requested (p. 1, line 24).

17. ("Page 2, Line 2: Surface melt is not restricted to the ablation zone but occur in the accumulation zone too.")

**Reply:** "across the ablation zone of" has been replaced by "on" (p. 2, line 8).

18. ("Line 3: "Greenland ice surface" should be "Greenland ice sheet"")

**Reply:** Changed as requested (p. 2, line 9).

19. ("Line 3: "can be" should be "is"")

**Reply:** Changed as requested (p. 2, line 10).

20. ("Line 14: Bartholomew et al. (2011) does not seem to be a fitting citation here as this paper treats of observations.")

**Reply:** Bartholomew et al. (2011) has been deleted, as requested (p. 2, line 21).

21. ("Page 3, Line 3: "to discern", to is missing")

**Reply:** "to" has been added, as requested (p. 3, line 19).

22. ("Page 4, Line 13: The parameters Cp and Ct should be explained.")

**Reply:** $C_p$ and $C_t$ are two parameters depending on "units and drainage-basin characteristics". This has been explained, as requested (p. 5, lines 28-29).

23. ("Line 13: "time-to-peak in" reads strangely.")

**Reply:** "in" has been deleted (p. 5, line 28).

24. ("Line 15: I am not sure that the citations are needed here an interested reader will find those in Smith et al. (2017)")

**Reply:** These two citations have been deleted, as requested (p. 5, line 31).

25. ("Page 5, Equation 2: t; tc and th are not described in the text.")

**Reply:** $t_h$ is the hillslope travel time and $t_c$ is the channel travel time. This has been explained, as requested (p. 7, line 2).

26. ("Line 16: Replace "research" by "study".")

**Reply:** Changed as requested (p. 4, line 8).

27. ("Page 6, Line 6: The contributing area (Ac) should be introduced and discussed in the model description.")

**Reply:** Cumulative contributing area ($A_c$) defines the surface area needed to initiate open channel flow. Larger $A_c$ values will yield smaller open-channel zones because larger contributing interfluve areas are required to form open channels. Additional new text has been added to explain $A_c$, as requested (p. 8, lines 15-16, 20-21).

28. ("Line 15: "compute" rather than "derive".")

**Reply:** Changed as requested (p. 8, line 24).

29. ("Line 19: "framework" could be omitted.")

**Reply:** Changed as requested (p. 8, line 29).

30. ("Page 7, Line 6: "climate model" can be skipped here.")

**Reply:** Changed as requested (p. 9, line 20).

31. ("Line 7: The times given here do not agree with the one that are present on Figure 3. The author should choose which are the more relevant and keep them throughout.")

**Reply:** The peak discharge time 13:00-15:00 is shown in Table 2 rather than Figure 3. This point has been better explained, as requested (p. 9, line 22).

32. ("Line 16: I don't agree with the statement on the smoothness of the UHs. From the figure it seems that the UHs from SUH are actually the smoothest of all.")

**Reply:** We mean the RWF UH is flatter than SUH and 2 m SRLF UH because its peak UH value is smallest, thus distributing surface meltwater more 'smoothly' over time. We now think this is confusing as the reviewer pointed out so we have changed this sentence into "The peak values of RWF UHs are smaller than SUHs and 2 m SRLF UHs therefore RWF UHs temporally distribute surface meltwater most smoothly", as requested (p. 10, lines 4-5).

33. ("Page 8, Line 2: Shouldn't it be "potential dynamism"?")

**Reply:** Changed as requested (p. 12, line 5).

34. ("Page 9, Line 16: Figure 7 actually shows the results from the three different models not only SUH. The comparison between the results of SHAKTI with the forcing from the RCM and the various models should be presented here to convince the reader of the advantage to use those models. As stated before, the setup of the subglacial hydrology model should be corrected to give convincing results. I am also unsure of the location where the effective pressure presented on Figure 7 is sampled from the model.")

**Reply:** In the original manuscript, the effective pressure was sampled from the location on the bed where the meltwater is input (i.e. the "moulin location" on the bed). Similar behavior is seen by examination of the mean effective pressure instead, however, and we have altered the revised manuscript to focus on this quantity (see our reply to your comment 11).

35. ("Line 30: The study from Chandler et al. (2013) actually shows subglacial travel time. I don't see how this reference fits here.")

**Reply:** Chandler et al. (2013) focused on subglacial travel time, as the reviewer pointed out but also reported peak supraglacial river discharge time for an IDC at southwest GrIS (moulin site L41 in their study) during 29 June to 7 July 2011. Thereby, we suggest that this reference fits here.

36. ("Page 10, Line 18: Should be "bare ice".")

**Reply:** Changed as requested (p. 16, line 16).

37. ("Figure 2: I don't think that Figure 2 is necessary and it could be skipped.")

**Reply:** Figure 2 has been deleted, as requested.

38. ("Figure 3: This figure is quite hard to read as it holds a lot of information. I would suggest to plot on this figure only the optimal simulations for SRLF and RWF which would allow an easier and more fair inter-comparison of the models. Another solution might be to split the figure to present the inter-comparison on a specific figure and the sensitivity studies on others. Finally, a zoom on some relevant period for the discharge and effective pressure would help the comparison of the different models. I also noticed a discrepancy here between the times given in the first column and the one of the text. It would be advantageous to introduce the RCM instantaneous runoff in the first column for ease of comparison.")

**Reply:** We have revised Figure 3 based on your and Reviewer 2's comments (p. 24). See our reply to your comment 10.

Figure 7 shows the average two-day cycle of moulin discharge (Q) for IDC1 during July 2015 derived from Figure 3. As such, Figure 7 is "a zoom on some relevant period" as the reviewer suggested. We have better illustrated Figure 7 to compare different routing models: "A magnified example of this timing is seen in Figure 7, which presents the average two-day cycle of moulin discharge input using the three routing models overlaid with effective pressure in IDC1. All three routing models achieve minimum moulin discharges around 09:00-11:00 and minimum effective pressures around 17:00-19:00, yielding a time lag of 8-9 hours; in contrast, the RCM instantaneous runoff without routing achieves minimum moulin discharge around 00:00 and minimum effective pressure around 10:00. The timing of effective pressure produced using RCM instantaneous runoff is visibly different than with the routing methods; interestingly, the timing of minimum effective pressure simulated by the RCM instantaneous runoff is very close (~ 1 hour) to that of maximum effective pressure simulated by the routing models", as requested (p. 27).

39. ("Figure 4: Figure four is barely described in the text, it should either be better described or completely omitted.")

**Reply:** Figure 4 shows scatter plots of RWF-routed moulin diurnal discharge range (difference between maximum and minimum moulin discharge) vs. those modeled from RCM instantaneous runoff, SUH routing, and SRLF routing. We now think it is not closely related with the main topic of this study so we have deleted it, as suggested.

40. ("Figure 5: Ac is given here in km2, it should be given in m2 for consistency with the rest of the manuscript. The caption here could be shortened to its descriptive part.")

**Reply:** The area unit has been changed into "$m^2$" and the caption has been shortened, as requested (Figure S5).

[Figure]

**Figure S5.** Variable supraglacial stream/river network for IDC1, as simulated by applying variable accumulative area threshold (Ac) values to ArcticDEM.

41. ("Figure 7: As for Figure 3 this figure is quite busy and should be simplified. The caption here is not adequate with some description missing and some discussion points that could be stripped.")

**Reply:** Figure 7 has been simplified and better explained, as requested (p. 27). See our reply to your comment 10.

42. ("References: dois are missing from the references")

**Reply:** DOIs have been added, as requested (p. 19-22).

**Reply:** We suggest that the three models (SUH, RWF, and SRLF) are all routing models. A routing model does not need to explicitly determine how meltwater produced on a cell is routed to its downstream cell(s) in a catchment, which is the aim of spatially-distributed routing models. In contrast, a routing model can be lumped and only determines how

surface meltwater produced in the catchment is temporally routed to the catchment outlet. Unit hydrograph (UH) is designed for this purpose. UH is a transfer function that models catchment runoff response to rainfall (melt in our case) events for some unit duration and unit depth of effective water input. In this study, all the three models (SUH, RWF, and SRLF) exhibit their UHs and consequently can temporally route surface meltwater to the catchment outlet and yield moulin discharge hydrographs. From this perspective, we suggest that they are all routing models. This point has better explained (p. 5, lines 20-23).

Moreover, RWF and SRLF actually work very similarly. SRLF uses Manning's open-channel equation to calculate meltwater flow velocity for each cell of a catchment, while RWF uses constant hillslope and open-channel flow velocities calibrated from field measurements to determine each cell's flow velocity. As such, RWF and SRLF both generate a velocity raster. Then, integrating velocity with flow path distance calculated from DEM, meltwater transport time from each cell to the catchment outlet can be determined and the transport time distribution yields UH. In short, the primary difference between RWF and SRLF is the way they determine flow velocity for each cell.

We have added Table S1 to summarize the benefits of using different routing models and better illustrated the three routing models in the methods and data section (p. 4, lines 2-15; p. 5, lines 13-31; p. 6, lines 1-16; p. 7, lines 1-13; p. 8, lines 2-8).

**Table S1. A brief summarization of surface meltwater routing models.**

| Model | Meltwater Routing | Applicable on bare ice surfaces | | Applicable on snow surfaces | Parameter dependency | DEM dependency | Case study |
|---|---|---|---|---|---|---|---|
| | | Hillslope | Open-Channel | | | | |
| **Instantaneous RCM runoff** | No | - | - | Yes | No | No | (McGrath et al., 2011; Bartholomew et al., 2012; Rennermalm et al., 2013; Fitzpatrick et al., 2014) |
| **Snyder Synthetic Unit Hydrograph (SUH)** | Yes | - | - | Yes, but model parameters should be recalibrated | $C_p$, $C_t$ are calibrated using a field-measured moulin hydrograph | No | (Smith et al., 2017) |
| **Surface Routing and Lake Filling (SRLF)** | Yes | No | Yes | Yes | No | DEM is required to calculate meltwater flow velocities for all catchment cells | (Arnold et al., 1998; Willis et al., 2002; Banwell et al., 2012; Banwell et al., 2013; Arnold et al., 2014; Banwell et al., 2016; de Fleurian et al., 2016; Koziol and Arnold, 2018) |
| **Rescaled Width Function (RWF)** | Yes | Yes | Yes | Yes, but model parameters should be recalibrated | Hillslope and open-channel flow velocities ($v_h$ and $v_c$) are calibrated using a field-measured moulin hydrograph | High-resolution (<10 m) DEM is required to calculate hillslope flow path length | (Yang et al., 2018) |

3. ("Introduction: In general, I do not find that the introduction sets up the objectives of the paper very well. It does not provide sufficient information to set up a methods comparison, but also only emphasizes the subglacial channel evolution at the end – like an afterthought. I suggest the following changes: - You need to be clearer in your introduction that this is not a methodological paper per se – as you say later in the paper, you cannot say which method performs better due to a lack of empirical evidence. You can only speculate on the modelled (not observed) hydrological implications of the three different methods. This needs to come across more strongly.")

**Reply:** We have included the following statement in the Introduction: "Notably, this study cannot, in good faith, focus on method comparison, nor determine the 'best' (i.e., best able to reproduce a real-world moulin hydrograph) due to the lack of calibration and validation data. Owing to this limitation, the goal of this study is to assess differences among the three meltwater routing models, rather than revealing which model most realistically simulates surface meltwater routing on the ice surface. By using the outputs from all three as meltwater inputs to drive the SHAKTI subglacial model, we characterize the impact of their differences on subglacial effective pressure, and, more generally, the importance of routing supraglacial runoff on subglacial conditions", as requested (p. 4, lines 6-15).

4. ("The differences between the three approaches as well as the assumptions and limitations of each needs to be made clearer for a glaciology (rather than a hydrology) readership. The limitations of the empirically-derived RWF and SUH also need to be made clear, particularly the temporal limitations of the original field-derived moulin hydrograph measurement. As is, a comparison of these three approaches is only useful for conditions similar to those under which the SUH and RWF approaches are calibrated – this needs to be made clear.")

**Reply:** We have better explained the three approaches and made them clearer for the glaciology readership, including the addition of Table S1. The limitations of the empirically-derived RWF and SUH have been made clearer, as requested (p. 14, lines 6-13).

SUH and RWF both rely on several empirically parameters ($C_p$ and $C_t$ for SUH, and $v_h$ and $v_c$ for RWF) calibrated from a moulin hydrograph measured at the Rio Behar catchment, southwest GrIS during a very short time period (72 hours), July 2015 (Smith et al., 2017). In contrast, SRLF is more solid because it only relies on DEM to calculate meltwater flow velocities (Banwell et al., 2012). In this study, we assume these empirically parameters are transferable over space and time but this assumption needs further validation. It may hold for ice sheet surface with similar hydrologic and glaciological environments but it may be problematic to apply over larger space and longer time. A second independent, long-term moulin hydrograph will help to address this problem.

5. ("Because the comparison between the approaches is inherently limited due to the temporally-limited nature of the field moulin discharge measurements in the SUH and RWF, I think it would be useful in the introduction to put more emphasis on the goal of the paper as an exercise in examining (modelled) subglacial effective pressure sensitivity to diurnal

hydrographs, rather than explicitly a comparison of moulin discharge estimate approaches. I would consider a bolder introductory statement around Page 2, Line 17 that frames the paper as a (preliminary) investigation of the extent to which moulin hydrograph estimates matter for affecting modelled subglacial hydrology and effective pressures.")

**Reply:** We have included two modifications to the introduction to emphasize the importance of the supraglacial system on subglacial hydrology. First, the current lack of representation of the surface meltwater routing leads to an insufficient understanding of surface-to-bed meltwater connections and ice dynamics, particularly on diurnal timescales. Therefore, constraints on IDC discharge can provide critical boundary conditions for studies of the subglacial hydrologic system (p. 2, lines 28-30). Second, by using the outputs from all three as meltwater inputs to drive the SHAKTI subglacial model, we have characterized the impact of their differences on subglacial effective pressure, and, more generally, the importance of routing supraglacial runoff on subglacial conditions (p. 4, lines 2-5). We have also emphasized the subglacial results in the last line of the abstract (p. 2, lines 3-6). We thoroughly discuss the implications of these different surface meltwater routing models on subglacial hydrology in section 4.2.

6. ("Last paragraph of introduction. I think there needs to be more explanation of what you are hoping to achieve with using the SHAKTI model. I think you need to be explicit that there is no objective away to compare the three moulin hydrograph methods, and the differences between them only matter in a glaciological sense if they significantly impact subglacial hydrology and effective pressure. You therefore run what is loosely a sensitivity test using SHAKTI to assess the modelled impacts of each approach. Although this comes out later in the paper, you need more framing of this consideration in the introduction.")

**Reply:** See our reply to your comment 3. We have revised the last paragraph of introduction to better illustrate the objective of this study and explicitly indicate our goals with the lines: "By using the outputs from all three routing models as meltwater inputs to drive the SHAKTI subglacial model, we seek to characterize the impact of differences in surface routing on subglacial pressures and evolution, particularly over diurnal timescales. More generally, these results can demonstrate the extent to which the choice of surface meltwater routing algorithms can alter modelled subglacial conditions", as requested (p. 4, lines 6-15).

7. ("Study area and datasets: More specific justification of the chosen study IDCs is needed. They are approximately similar sizes to the IDC used in the Smith et al. (2017) measured moulin hydrographs, which should be pointed out, and they also appear to exclude large supraglacial lakes, which is likely to affect the comparison of the SRLF approach with the RWF and SUH approach. This should be noted in the study area description and in the discussion.")

**Reply:** We have better explained the reasons to select the four IDCs: "They are distributed at approximately 200 m elevation intervals in order to span the elevational range of most well-developed IDCs found in the Russell Glacier region and the variable surface melt conditions of this region (Yang and Smith, 2016). Large supraglacial lakes are absent in these four IDCs

(Figure 1) and surface meltwater is all routed to the moulin at the catchment outlet. As such, surface runoff produced in each IDC equals to the moulin discharge (Smith et al., 2017). A moulin discharge hydrograph collected at Rio Behar catchment (IDC2 in this study; 67.049346N, 49.025809W) for 72 h from 20 to 23 July 2015 was used to calibrate parameters of SUH and RWF models (see Sections 2.5 and 2.6). It is problematic to apply these empirically-derived parameters over large spaces and long times (Yang et al., 2018). Therefore, the selected IDCs are distributed in a relatively small region and the areas of IDC1, IDC3, and IDC4 are similar to the Rio Behar catchment (IDC2)", as requested (p. 4, lines 20-31).

We have better illustrated the limitations of SUH and RWF in the discussion section, as requested: "SUH and RWF both rely on several empirically parameters ($C_p$ and $C_t$ for SUH, and $v_h$ and $v_c$ for RWF) calibrated from a moulin hydrograph measured at the Rio Behar catchment, southwest GrIS during a very short time period (72 hours), July 2015 (Smith et al., 2017). In contrast, SRLF is more solid and applicable over large spaces and long times because it only relies on DEM to calculate meltwater flow velocities (Banwell et al., 2012). In this study, we assume these empirical parameters are transferable over space and time but this assumption needs further validation. It may hold for ice sheet surface with similar hydrologic and glaciological environments but is problematic to apply over larger space and longer time. A second independent, long-term moulin hydrograph will help to address this problem" (p. 14, lines 6-13).

8. ("Methods: Overall, the methods seem written for hydrologists, not for glaciologists. More information is needed in this section to make it useful to its readers.")

**Reply:** We have better introduced the three routing models from a perspective of glaciologists. A new section has been added to explain Unit Hydrograph and its application for calculating moulin discharge. Additional new text has been added to better explain Snyder Synthetic Unit Hydrograph, Surface Routing and Lake Filling, and Rescaled Width Function, as requested (p. 4, lines 2-15; p. 5, lines 13-31; p. 6, lines 1-16; p. 7, lines 1-13; p. 8, lines 2-8).

9. ("Presumably, July 2015 was chosen for the MAR runoff simulations because that is coincident with the field-collection of the moulin discharge hydrograph. This should be made clear, so that the constraints on the method are obvious to readers.")

**Reply:** Yes, July 2015 runoffs were derived to be coincident with the field-collection of the moulin discharge hydrograph. Additional new text has been added to explain this point, as requested (p. 5, lines 8-9).

10. ("If this paper is to be a useful methodological resource for glacier hydrologists, a more complete comparison of the three approaches is needed. Perhaps a table would be useful in comparing the three methods – this table could keep track of the references, acronyms, assumptions, limitations, etc…")

**Reply:** Thanks for this great comment. We have added a new table (Table S1) to better compare the three routing models, as suggested (see our reply to your comment 2).

11. ("Section 3.6 – your explanation of a 'dynamic' Ac is not clear, or perhaps I have missed it. In Figure 3, it looks like you tested all five of the different Ac values independently over the whole time span, but then there is also a 'dynamic Ac'. Is a 'dynamic' Ac one that evolves according to your six five-day RWF-UHs? Be sure to call that 'dynamic' here.")

**Reply:** Yes, we tested all five of the different $A_c$ values independently over the whole time span and then a 'dynamic' $A_c$ one evolves according to the six five-day RWF-UHs, as the reviewer pointed out. We have added a new sentence "Each RWF UH was conducted to calculate moulin discharge for five days and the resultant moulin discharge is termed as dynamic $A_c$ discharge" to better explain dynamic $A_c$, as requested (p. 8, lines 20-21).

12. ("Results: the figures for this section are confusing. They need to be pulled apart to be more readable, and the legends and captions need better information.")

**Reply:** We have carefully revised the figures to make them more readable, as requested. See our reply to your following comments (p. 24-28).

13. ("Figure 3 is too confusing with this many panels, and the result is that the lines are too small to make out the subtle differences due to the different variables. I suggest taking the third column (effective pressure) out and putting it in its own figure, as it is a distinct part of the results and discussion. You could then make the main figure slightly larger, and show with a title on the legend in (g) and (j) that the different series refer to g) the DEM resolution and j) the channel initiation threshold (proxy for time).")

**Reply:** We have taken the third column (effective pressure) and put it in its own figure, as the reviewer suggested (Figures S1 and S2). We have made Figure 3 larger and reshaped the width ratio between the first column (UH) and the second column (moulin discharge) from 1:1 to 1:1.5 to better represent the diurnal moulin discharge. We have added a title "DEM resolution" to Figure 3c and 3e, and a title "Ac" to Figure 3g, changed the x-axis labels of Figure 3e and 3g into "RWF UH ($A_c$ = 100 m$^2$) and "RWF UH (2 m DEM), respectively, and added legend "RCM runoff" to Figure 3b, 3d, and 3f. We have better explained "the channel initiation threshold (proxy for time)" in the caption of Figure 3.

[Figure]

**Figure S1.** Presentation of Unit Hydrographs (UHs) (column 1) and moulin discharges (column 2) of IDC1 during July 2015, as simulated by three supraglacial routing models (SUH, SRLF, and RWF). $A_c$ is the cumulative contributing area required to initiate a supraglacial meltwater channel and dynamic $A_c$ values are used as proxy for time to simulate the temporal evolution of supraglacial stream/river networks. Simultaneous RCM runoff (grey line) is shown to indicate the effect of surface meltwater routing process on moulin discharge.

[Figure]

**Figure S2.** Effective pressures for IDC1 simulated by SHAKTI, with inputs to the subglacial system via a single moulin prescribed by the moulin discharges (shown in Figure S1) calculated by the various routing models. The effective pressure shown here is the spatial mean for the entire 1-km square domain which contains the moulin input at its center.

14. ("Figure 4 is not discussed in the results section. Some discussion should be provided in the 'long-term evolution' section, or it should be removed.")

**Reply:** We agree Figure 4 is not closely related with the main topic of this study so we have removed it, as suggested.

15. ("Figure 3k – how is that there is so much smaller discharge for 5000m$^2$ than there is for 100m$^2$? With a higher Ac, there should be less efficient routing, lower peak Q and a flatter hydrograph but still, presumably, similar discharge. This distinction is not clear from the figure, perhaps because the lines are so compressed and the 'flashiness' of the hydrographs is not clear. Perhaps an inset figure would be helpful.")

**Reply:** $A_c$ = 5000 m$^2$, as the reviewer pointed out, yields lower peak Q and a flatter hydrograph than $A_c$ = 100 m$^2$ but it also yields higher minimum Q. Therefore, the daily Q values calculated from $A_c$ = 5000 m$^2$ and $A_c$ = 100 m$^2$ are similar.

16. ("Discussion: Overall, I think this section is too critical of SRLF and not critical enough of SUH and RWF. Some discussion of the limitations of the latter two is needed, namely: the chosen study catchments do not appear to have lakes, and those methods may not perform adequately in catchments with lakes, at different times of year and in different snow/ice/ surface slope conditions than those in which the field-measurements of the moulin hydrograph (Smith 2017) were collected.")

**Reply:** SRLF is the first model to route surface meltwater on the Greenland Ice Sheet. We think it is very successful. The SRLF model employs Darcy's law to route surface meltwater flow through snow and Manning's open-channel flow equation to route meltwater flow over bare-ice surfaces (Arnold et al., 1998). SRLF has been applied to simulate supraglacial lake growth and to drive subglacial hydrological evolution during several entire melt seasons (Banwell et al., 2012; Banwell et al., 2013; Arnold et al., 2014; Banwell et al., 2016). In this study, we focus on its bare-ice part to make it comparable with the SUH and RWF because the coefficients of these two models were calibrated using a field-measured moulin hydrograph on bare ice surface (Smith et al., 2017; Yang et al., 2018). It may 'hurt' the full ability of SRLF model as the reviewer pointed out.

To address this problem, we have: (1) better introduced SRLF model in the methods and data section (p. 6, lines 8-16), and (2) better discussed the limitations of SUH and RWF as requested (p. 6, lines 3-6; p. 7, lines 9-12; p. 14, lines 6-13). We have illustrated the limitations of SUH and RWF in detail in our previous studies (Smith et al., 2017; Yang et al., 2017) so we have only illustrated their limitations when comparing with SRLF and driving subglacial hydrology.

17. ("More discussion of the limitations of SHAKTI would be helpful. You should be clear that SHAKTI is used to provide preliminary insight into the possible importance of accurately capturing the details of an hourly moulin hydrograph, and that many complexities are not

**Reply:** The limitations of SHAKTI itself are enumerated in Sommers et al. (2018). We have included a statement indicating the limitations of the model runs themselves in the results section: "The subglacial model domain and duration were chosen to illustrate the impact of the chosen supraglacial routing model on local subglacial hydrology in the vicinity of a moulin input at the bed. As such, our results cannot necessarily be extrapolated to infer large-scale or seasonal evolution of the subglacial hydrologic system in response to different surface forcings; however, the results do provide insight into the potential diurnal sensitivity of the subglacial system to changes in supraglacial meltwater routing and the associated modification of the discharge hydrograph" (p. 11, lines 20-32).

In the updated manuscript, we have rerun the SHAKTI simulations with more realistic boundary conditions, surface slopes, and sliding velocities for each catchment to better represent actual effective pressures that may be found in the vicinity of a moulin in each region. The reviewer is correct that this is a simple exploration of the influence of different surface meltwater methods on subglacial pressures, but we hope that it provides a view into these connections and may serve as inspiration and motivation for more detailed studies involving simulations of multiple catchments with multiple realistic moulin inputs, topography, etc. In terms of capturing complexities of the subglacial drainage system, even with these small-scale simulations, the SHAKTI model does realistically represent realistic flow and pressure regimes, and evolving geometry under the ice in the vicinity of a moulin.

Technical corrections: Please see specific in-text comments in the attached annotated PDF. Please also note the supplement to this comment:

18. ("P2, line 12, Flowers, 2018, I would drop this reference as you provide specific references in the next sentence.")

**Reply:** This reference has been deleted, as requested (p. 2, line 18).

19. ("P2, line 31, "field-measured moulin hydrograph", more information needed. Be clear that this is based on one moulin for one moment in time - make the limitations clear.)

**Reply:** The field-measured moulin hydrograph was collected at Rio Behar catchment, southwestern GrIS (67.049346N, 49.025809W) for 72 h from 20 to 23 July 2015. Although Smith et al. (2017) demonstrated coefficient transferability using two other independently field-measured moulin hydrographs, the two calibrated coefficients are collected at "one moulin for one moment in time" as the reviewer pointed out so they may still be limited to apply over longer time and larger areas. Additionally new text have been added to explain this point, as requested (p. 4, lines 27 -31).

20. ("P3, lines 3-4, "Catchment-averaged meltwater transport velocities for each zone were then calibrated using a field-measured moulin hydrograph (Smith et al., 2017)", More information needed. Be clear that this is based on one moulin for one moment in time - make the limitations clear.)

**Reply:** See our reply to your comment 19.

21. ("P3, line 6, "it only requires catchment shape and area to estimate surface meltwater transport time (Smith et al., 2017)", As well as a number of important empirically-derived parameters. That should be made clear here.")

**Reply:** Changed as requested (p. 3, line 23).

22. ("P4, line 5, "RCM runoff simulations", Avoid acronyms in the subtitles.")

**Reply:** This subtitle has been changed into "2.2 Regional climate model runoff simulations", as requested (p. 5, line 8).

23. ("P4, line 14, "field-measured moulin hydrograph", Specify where and when this data was collected.)

**Reply:** This data was collected at Rio Behar catchment, southwestern Greenland Ice Sheet (67.049346N, 49.025809W) for 72 h from 20 to 23 July 2015. Additional new text has been added to explain this point (p. 5, line 30).

24. ("P4, line 23, If you are going to supply the equation here, supply it for the relevant SUH equations above as well.)

**Reply:** The relevant SUH equations have been added, as requested (p. 5, lines 26-29).

25. ("P4, line 28, Spell this out here - it is difficult to keep track of all the acronyms.)

**Reply:** Changed as requested (p. 6, lines 24-25).

26. ("P5, lines 14-19, Make this its own section - at the moment, it is tucked under RWF. Alternatively, move it to the introduction or to section 3.7")

**Reply:** We have removed this paragraph to the introduction section, as suggested (p. 4, lines 6-15).

27. ("P6, lines 1-2, Is it possible that the empirical parameters Cp and Ct might change seasonally?)

**Reply:** Good point. $C_p$ and $C_t$ are two empirical parameters that quantify the hydrologic response of a catchment to surface melt. If we can have a moulin discharge hydrograph during the entire melt season, we can calculate multi-temporal $C_p$ and $C_t$ using variable time-to-peak ($t_p$), peak discharge ($h_p$), and main-stem stream length ($L$), thus creating multi-temporal SUHs. Without such direct measurements, the parameters cannot be realistically varied in time and SUH cannot mimic variable hydrologic response of a catchment to surface

melt. We have better explained this point in the Methods and Data section (p. 8, lines 5-8).

28. ("P7, line 8, I keep getting up on routing. These are not routing processes in the RWF and the SUH??")

**Reply:** See our reply to your comment 2.

29. ("P7, line 14, But of course this is the catchment from which the empirical parameters are derived, so is the comparison really appropriate? At least make it clear.")

**Reply:** We agree with the reviewer that this comparison is limited since we do not have a second field-measured moulin discharge hydrograph as validation data. But we suggest that this comparison at least indicates the transferability of these empirical parameters ($C_p$ and $C_t$ for SUH, and $v_c$ and $v_h$ for RWF) from the field-measured catchment to other catchments with similar areas. The limitation of the comparison has been better explained, as requested (p. 9, line 29; p. 14, lines 6-13).

30. ("P7, line 28, Is this really 'long term'? 'Temporal evolution' would be better.")

**Reply:** We have changed 'long-term' into 'temporal', as suggested (p. 12, line 1).

31. ("P11, line 17, Should be decreases because subglacial water pressure will increase?")

**Reply:** Good catch! Changed and clarified to: effective pressure decreases, resulting in increased sliding velocities (p. 14, line 21).

32. ("P21, Figure 3, Remove the 'UH' - it looks as though you are subtracting a function from a function. With this legend down here, it is not clear that the grey refers to all plots. Is only 100 and 5000m2 showed in this figure? If so, it should say so in the legend.)

**Reply:** 'UH' has been removed, as requested. The grey refers to all plots and moulin discharge hydrographs for all six cumulative area thresholds are showed in Figure 3k. We have revised Figure 3 to make it more understandable (p. 24-25).

33. ("P22, Figure 4, This figure is not explained in text and its context is therefore not clear")

**Reply:** We now think Figure 4 is not closely related with the main topic of this study so we have removed it, as suggested.

34. ("P25, Figure 7, Why are there two solid lines and two dashed lines in each? What do the grey lines mean? Are they RCM discharges?")

**Reply:** Two solid lines are supraglacial moulin discharge and two dashed lines are effective pressure. The grey solid lines are RCM surface runoff and the grey dashed lines are effective

pressure simulated from RCM surface runoff. All the line colors correspond with the ones in Figure 3. This point has been better explained. We have updated Figure 7 to make it more understandable (Figure S3).

The average two-day cycle is presented because the average one-day cycle cannot present complete results for all models. All the three routing models achieve minimum moulin discharges around 09:00-11:00 and minimum effective pressures around 17:00-19:00, yielding a time lag of 8-9 hours; in contrast, the RCM instantaneous runoff without routing achieves minimum moulin discharge around 00:00 and minimum effective pressure around 10:00 (Figure S3). All three routing models achieve minimum moulin discharges around 09:00-11:00 and minimum effective pressures around 17:00-19:00, yielding a time lag of 8-9 hours; in contrast, the RCM instantaneous runoff without routing achieves minimum moulin discharge around 00:00 and minimum effective pressure around 10:00. The timing of effective pressure produced using RCM instantaneous runoff is visibly different than with the routing methods; interestingly, the timing of minimum effective pressure simulated by the RCM instantaneous runoff is very close (~ 1 hour) to that of maximum effective pressure simulated by the routing models.

[Figure]

**Figure S3.** The average two-day cycle of moulin discharge (Q) for IDC1 during July 2015. The daily minimum input in supraglacial moulin discharge (solid lines) corresponds generally to maximum effective pressure (dashed lines), and is followed within 8-9 hours by the daily minimum effective pressure (maximum subglacial water pressure). This suggests that the system shuts down due to creep with low meltwater input, and becomes highly pressurized as meltwater input increases again. As the new water inputs are accommodated, efficient pathways reform and effective pressure increases (subglacial water pressure decreases).

Thank you for considering this manuscript for publication in *The Cryosphere*. If we may provide any additional information about the dataset or analysis, please do not hesitate to contact us at yangkangnju@gmail.com.

Respectfully submitted,

Kang Yang
Associate Professor

[revised manuscript text omitted]

In general, we find that while the moulin inputs generated by the different surface routing models produce variations in diurnal range of effective pressure, the overall channelization behaviour and temporal mean effective pressure over the 31-day simulations are relatively consistent between models.  The subglacial model domain and duration were chosen to illustrate the impact of the chosen supraglacial routing model on local subglacial hydrology in the vicinity of a moulin input at the bed. As such, our results cannot necessarily be extrapolated to infer large-scale or seasonal evolution of the subglacial hydrologic system in response to different surface forcings; however, the results do provide insight into the potential diurnal sensitivity of the subglacial system to changes in supraglacial meltwater routing and the associated modification of the discharge hydrograph. For example, the amplitude of diurnal effective pressure variation for a particular day may range from < 0.5 MPa to > 3.0 MPa, depending on the surface method used (Figure 3). While future work should pursue broader-scope simulations to more fully investigate the larger- and longer-scale effects of surface input variations on effective pressure and ice dynamics, our results suggest that in a fully coupled ice dynamics/subglacial hydrology model, different routing methods may not produce significantly different cumulative or time-averaged effects in effective pressure for simulation time scales longer than daily.

**3.3 Temporal evolution of moulin discharge simulations**

 To illustrate the strong potential dynamism of supraglacial stream/river network extent on bare ice, Figure 6 presents dynamic supraglacial stream/river networks of IDC1 under different channel initial $A_c$ thresholds (proxy for time). Numerous small supraglacial streams are generated when $A_c$ is set to 5000 m$^2$ (Figure 6a), while only large main stem channels are obtained when $A_c$ is set to $10^6$ m$^2$ (Figure 6d), yielding drainage density varying from 19.3 m$^{-1}$ to 0.8 m$^{-1}$. Such strong, seasonal variation in supraglacial stream/river network extent is well-supported by previous high-resolution remote sensing studies, which report similar expansion and contraction of actively flowing supraglacial stream/river networks during the melt season (Smith et al., 2015;Smith et al., 2017;Yang et al., 2018).

Well-developed supraglacial stream/river networks (e.g., simulated using $A_c = 100$ m$^2$) route surface meltwater efficiently (Smith et al., 2015), yielding high UH and moulin discharge peaks, whereas relatively poorly-developed supraglacial stream/river networks indicate relatively slow routing of meltwater in expanding interfluve zones (Yang et al., 2018), yielding smaller and delayed UH and moulin discharge peaks (Figures 2g and 2h). As such, temporal evolution of supraglacial stream/river networks substantially alter surface meltwater routing processes and yield dynamic unit hydrographs and moulin discharges. This result is consistent with Montgomery and Foufoula-Georgiou (1993) and van Meerveld et al. (2019), which found that river network extent controls the response of catchments to precipitation (melt in our case). During the first 10 days (240 hours), the channel initiation thresholds $A_c$ are small so numerous supraglacial streams/rivers are developed; thereby, the dynamic $A_c$ RWF routing performs similarly with SUH, 2 m SRLF, and the original RWF. As supraglacial stream/river networks shrink (i.e., $A_c$ becomes larger), our simple modelling experiment (by increasing $A_c$ thresholds every 5 days for the month of July 2015) suggests a gradual dampening of diurnal variations until, by the end of the month, such variations are ~50 % smaller than the same routing method using a smaller $A_c$ value (Figure 2h). Moreover, the average two-day cycle of moulin discharge simulated by the dynamic $A_c$ RWF routing is considerably dampened compared to those simulated by SUH, 2 m SRLF, and the original RWF (Figure 6). In the dynamic $A_c$ RWF routing, the dampening effect displayed in the moulin discharge inputs carries over into the resulting effective pressure cycles as well (Figures 3 and 6).

**3.4 Effects of DEM spatial resolution on surface meltwater routing**

RWF routing is largely unaffected by DEM spatial resolution, but SRLF routing is significantly affected. Resampling 2 m ArcticDEM data to 5 m and 10 m yields similar RWF UHs and moulin discharges (Figures 2e and 2f), but progressively smoother versions using SRLF UHs (Figures 2c and 2d). At 90 m resolution, SRLF UH shapes exhibit diminished and

delayed peaks (Figure 2c). For IDC1, for example, use of a 90 m DEM yields a ~0.10 UH peak at hour 9 and a ~0.04 UH peak at hour 17, whereas 2 m DEM yield a ~0.15 UH peak at hour 5 with all meltwater evacuated within 15 hours.

SRLF applied on higher-resolution DEMs yields larger peak moulin discharges and diurnal discharge signals (Figure 2d). For IDC1, SRLF-routed peak moulin discharge derived from 2 m DEM is 52.4 % larger than that from 90 m DEM, while the corresponding value for SRLF-routed diurnal discharge range is 179.0 %. This finding indicates that SRLF performance is strongly controlled by DEM spatial resolution, with coarser resolutions resulting increased damping of simulated moulin hydrographs. The dampening effect of SRLF moulin discharge induced by DEM spatial resolutions carries over into the resulting effective pressure cycles as well (Figure 3b), urging caution when using coarse-resolution DEMs to route surface meltwater and to simulate subglacial effective pressure.

**4 Discussion**

**4.1 Implications of surface meltwater routing method inter-comparison**

This study  conducts an inter-comparison of three surface meltwater routing models (SUH, SRLF, and RWF). Due to the lack of field-measured moulin hydrographs, it is difficult to determine which surface meltwater routing model performs most realistically over large areas and long times. However, our simulations of moulin discharges (Figures 2, and S1-S4) can provide some insightful information to qualitatively evaluate the performance of certain routing methods. First, inclusion of surface meltwater routing introduces lower peak moulin discharges and delayed time to peak relative to the RCM instantaneous runoff. Second, for IDCs with similar areas and elevations as those examined here, simulated peak moulin discharge consistently occurs between 19:30 and 22:00 (Chandler et al., 2013;Smith et al., 2017). The timing of peak discharge suggests that several routing scenarios are unlikely to be realistic, including any technique that uses unmodified RCM outputs  as  these values qualitatively underestimate peak discharge time.

Additionally, we can use observations of channelized subglacial pressure to strengthen our qualitative comparison. While exact field measurements are not available for direct comparison to our modelled effective pressures, limited field observations indicate that hydraulic head within subglacial channels varies diurnally by at least 40 m and up to 150 m in regions with slightly thinner ice (Cowton et al., 2013;Meierbachtol et al., 2013;Andrews et al., 2014). Furthermore, these reported pressure variations in channelized regions do not fall below 40% of overburden at the observed points. While somewhat influenced by the particular geometry, moulin inputs, and other parameters used,  our model results for effective pressure in the vicinity of a moulin show modelled mean effective pressure variations ranging from ~2 MPa to ~4 MPa for IDC1 (~26% to 50% of overburden pressure). The RCM runoff produces effective pressures with the largest amplitude fluctuations, and is

[revised manuscript text omitted]

---

## Referee Report (RR1)

**Review of the article entitled "Inter-comparison of surface meltwater routing models for the Greenland Ice Sheet and influence on subglacial effective pressures."**

**1 General comments**

This study focuses on different ways of treating the supraglacial drainage of water at the surface of ice sheets. The region of interest studied here is Russell glacier in South West Greenland. The authors presents an intercomparison of three different surface routing models and compare their results to the output of a Regional Climate Model (RCM). The different inputs are further compared through by using them as the forcing quantity provided to a subglacial drainage model. The conclusion of the study are that the use of a supraglacial drainage system allows to get a better representations of the lag of the water input to the moulins. Some sensitivity among models also allow to quantify the impact of the Digital Elevation Model resolution on the drainage characteristics

This study provides an interesting insight into the differences that arise from the use of different supraglacial drainage model. This new version of the manuscript presents some improvement with respect to the first iteration but it feels that some work still needs to be done to convey the full potential of the study. Improvements in the text increased the readability and make it more accessible for the glaciological community. The figures could be further improved to help with the comparison and present a more in depth intercomparison, in particular, the figures for all the IDCs should be included in supplementary if they are not provided into the text. The improvement to the subglacial hydrology set-up is definitely yielding better results but I still think that the impact of the boundary conditions (BC) on the system is quite prevalent, the set-up should be refined further or the impact of the BC should be clearly pointed in the text. Below are some more details and other points that should be considered.

**1.1 Abstract**

The abstract is now far easier to read and the work on that has paid off.

- **Page 1**

- Line 15: I would replace "and impact" by "where it impacts".

- Line 25: Shouldn't it be "indicate that"?

- Line 33. I have a hard time understanding the sentence starting here.

**1.2   Introduction**

The Introduction now gives a better understanding of the problem at hand and of the state of the art. One point that should still be improved is the description of the SRLF model, it should be stated that this model is able to treat both bare ice and snow covered regions but that only the bare ice component has been used in this study.

- **Page 2**

- Line 4: Greenland Ice Sheet acronym (GrIS) should be introduced here.

- Line 13: RCM can be omitted as it is already defined in the abstract.

- Line 13: the Sentence starting on this line "In these instances ..." is very long. It could beneficiate from rewording. The citation used to defined IDCs could be replaced by an original sentence here.

- Line 18: Is "very long term" accurate here considering the fact that the conclusion are pointing toward the fact that the routing scheme does not have a huge influence on timescales longer than a day.

- Line 28: As for the IDC, having the definition of the UH as a citation makes the sentence very long and heavy. Please consider to have those definitions as stand alone sentences.

- **Page 3**

- Line 16: "and have different data requirements".

- Line 29: I am not sure of the meaning of intra-compared here.

- Line 32: "demonstrate" seems to be an overstatement.

- **Page 4**

- Line 1: You mean, best model, approach?

**1.3   Methods and Data**

This section as been greatly improved but there are still some imprecision.

Page 4, Line 14, it is stated that there are no large lakes in the IDCs, I suspect that the reason is that the IDCs have been chosen to avoid the presence of lake which would hinder the results of some routing models. If that is so this should be explicited in the text.

Page 4, Line 27 and following, it seems that MAR is referenced twice. The second description starting line 30 should be kept but it should probably be introduced in the section above rather than have its own section.

Page 4, Line 30 and other places in the manuscript. I feel that the simplification of "surface runoff" to "$(R)$" is not needed and that the former should be kept throughout the text.

Page 7, Line 14, here it is suggested that SRLF routing scheme does not evolve throughout the melt season which is somewhat misleading. I agree on the statement that the drainage on the bare ice component does not evolve but if one uses the full model then the evolution from snow covered to bare ice terrain will lead to modifications in the seasonal draining pattern.

- **Page 5**

- Line 15: SUH have been introduced before and should not appear here.

- Line 15: "assumes" should be replaced by "assumes that".

- Line 18: I am not confident with the Snyder SUH but it seems strange to have a discharge expressed as inverse of time.

- Line 18: Equations should not be written in line to help clarity.

- Line 20: "from the 20 to the 23..."

- **Page 6**

- Line 2 and Table 1:SRLF was not used in de Fleurian et al. (2016), the basic of the routing is the same but the scheme use in this study way simplified. Hence I don't think that de Fleurian et al. (2016) has its place here.

- Equation 2: This equation would read better in its fraction form.

- Line 12: "synthetic unit hydrograph" and "rescaled width function" could be abbreviated here.

- Line 16: The acronym has been introduced before and is not necessary here.

- Line 30: "hydologic" should be replaced by "hydrological".

- **Page 7**

- Line 12: Is it really the transport distance that change or the type of drainage. This could use some clarification.

- Line 26: This sentence is repeated in the next one and could be omitted.

- Line 31: I am not sure that "conducted" is the good term here.

**1.4   Results**

The new presentation of the results in the manuscript improved the readability. However, the figures are still on the small side to be really able to compare the results of the different models which would be the main focus of an intercomparison paper. My concern on the effect of the boundary conditions on the effective pressure has not been settled by the new set-up, I would recommend to use a far larger domain to avoid these boundary effect even if only a small region of this domain is analysed later on. For example using the SHMIP geometry from de Fleurian et al. (2018) with a single moulin located at a convenient position could be an option. If the author decide that the present set-up is enough for their use I would expect a more in depth analyses of the results in light of the potential impact that the BC have on those. The impact of the BC is clearly seen on the animation. The water head is lower at the right end of the domain when the water input is low and the gradient is very high when the influx is high. I suspect that this impacts the amplitude of the effective pressure variations.

For the figure that are presenting results on the IDCs, all the IDCs should be presented. I agree that only IDC 1 should be discussed and presented in the paper itself but the other IDCs should be shown in supplementary. Also take car when submitting your supplementary figures that they have the same layout as the one of the main manuscript.

Regarding the layout of the figures I would suggest to make a clearer separation between the figures related to the intercomparison (Figure 2 a-f, Figure 5 a-c) and the one more related to the temporal evolution of moulin discharge. That would allow to focus on shorter time scales (a few days) on the intercomparison figures and allow an easier comparison for the reader. Bellow are my suggestion figure by figure.

- Figure 2: I suggest to remove panels g and h of this figure and have them as a separate figure. With those panels removed, the temporal evolution on the discharge is not as relevant as it seems to be quite similar throughout the season (at least the difference between model does not seem to evolve). Those discharge plot can then be focused to span just a few days which would ease the comparison between models. The supplementary figures for the other IDCs should be the same as the figure presented in the paper (again to ease comparison)

- Figure 3: I feel that this figure is only relevant for the dynamic $A_c$ simulations. For the other simulation the seasonal evolution is not as interesting and the intercomparison is easier to perform on Figure 5. I would Use Figure 5 (with the modifications bellow) here and move this one to supplementary.

- Figure 4: This figure should be shown for the other IDCs in the supplementary along with animations for all IDCs.

- Figure 5: To my eyes this figure is the one that really allow to compare the models. The comparison would be even easier if there was one panel used for the RCM results and then one for each model (presented in a single column). The results from RWF should probably be omitted here as the Moulin influx at this date is quite different form the other and can not really be compared. I also note that on this figure the panel are not subscripted and that it is not reproduced for the other IDCs in the supplementary.

- **Page 8**

- Line 30: "study" should be replaced by "studied"

- **Page 9**

- Line 10: UH is already introduced at this point and should be used here.

- Line 11: "cm" should be spelled out.

- Line 12: The sentence starting here is confusing. Perhaps state first that the results of the IDCs are similar before pointing to the figures.

- Line 18: I still think that smoother is not the correct term to use here, I would prefer "UHs with lower amplitude" for example.

- Line 20: Are the peak really higher than 25% higher or just around 25% higher, in the second case you should use $\sim 25\%$.

- Line 21: The comment above applies to the discharge range too.

- Line 23: See comments above with respect to inequality notations.

- **Page 10**

- Line 10: A reference to the figure is missing here.

- Line 12: "...surrounding bed perpendicular to flow." I don't see the meaning of that.

- **Page 11**

- Line 5: I don't see how the $A_c$ threshold with unit of area is a proxy for time.

- Line 24: It does not seem that it is Figure 6 that should be referenced here.

- Line 29: It should be said here that RWF can not use DEM of coarser resolution. As it is now this comparison seems to be a bit unfair towards SRLF.

- Line 30: As stated before I don't think that smoother is a good term here. It seems that "lower amplitude UH" would be more descriptive of what is seen.

**1.5 Discussion and Conclusion**

If the author decide to keep the current set-up for the subglacial hydrology model (which I do not recommend) a careful assessment of the effect of the Boundary conditions on the effective pressure variations should be given here. In particular when describing the amplitude of the effective pressure variations.

- **Page 12**

- Line 11: This title seems strange.

- **Page 13**

- Line 1: It should be "empirical"

- Line 4: It should be "empirical"

- Line 13: This sentence should be written the other way around, it is the variation in subglacial pressure that trigger variations in ice velocity.

- line 17: This is an important and interesting statement that could be investigated further. The Dynamic $A_c$ simulation gives a good test case to look into this hypothesis.

- Line 28: I don't think that the storage capacity of the supraglacial system is demonstrated, it is more presented.

- Line 31: This could be seen the other way around too and routing models could help to parameterise the storage term built into subglacial hydrology models. It would be interested to test that to see if they have exactly the same effect on subglacial water pressure.

- **Page 14**

- Line 5: It should also be stated here that RWF can only operate on High resolution DEMs.

**References**

de Fleurian, B., Morlighem, M., Seroussi, H., Rignot, E., van den Broeke, M. R., Munneke, P. K., ... Tedstone, A. J. (2016, OCT). A modeling study of the effect of runoff variability on the effective pressure beneath Russell Glacier, West Greenland. *J. Geophys. Res*, *121*(10). doi: 10.1002/2016JF003842

de Fleurian, B., Werder, M. A., Beyer, S., Brinkerhoff, D. J., Delaney, I., Dow, C. F., ... et al. (2018). Shmip the subglacial hydrology model intercomparison project. *J. Glaciol.*, 1–20. doi: 10.1017/jog.2018.78

---

## Referee Report (RR2)

**Review of revised submission of: "Inter-comparison of surface meltwater routing models for the Greenland Ice Sheet and influence on subglacial effective pressures" by Yang et al.**

The manuscript is much improved by the changes made, in particular the clarification of the figures, the addition of Table 1, and the clarification around the methods. I have one lingering comment with respect to the description of the methods: SRLF partitions flow into open-channel flow and flow through snow using Darcy's law, whereas the RWF partitions flow into interfluve or open-channel flow. However, in this study the SRLF assumes flow over the whole ice surface can be modelled using open channel flow, but the RWF continues to partition flow into 'fluve' and inter-fluve. The difference, as far as I understand, is that a hillslope can be snow free and therefore not appropriately modelled by Darcy's law, however it is also not appropriate to model a hillslope as a 'fluve' because flow processes differ there. Therefore, the RWF allows hillslopes and channels to be modelled separately even when they are bare ice, and the SRLF does not. I think a short note on this distinction would waylay any confusion for readers, as well as a very short explanation of how processes differ between fluve and inter-fluve regions even under bare ice conditions (presumably diffusive vs. advective properties, but perhaps there is more?). This will help the glaciological audience understand the distinction between 'hillslope/inter-fluve' flow and flow through the snow pack. This would also help readers digest section 4.5.

Otherwise, I recommend that the paper be published with the technical corrections outlined below.

- Spacing in the in-text references needs to be corrected (there needs to be space after, not before, the semi-colon).
- Spacing – page 3, line 29 (.Routing)
- Page 3, line 16: A in-text reference could be provided to Table 1.
- Page 4 - Heading 2.2 needs to be on a separate line
- Page 4, line 31 – an extra 'to be' needs to be removed.
- Page 5, line 18 – $hp$ needs to be corrected to $h_p$.
- Page 5, line 28 – perhaps unimportant, but I dislike the phrase "used to fill supraglacial lakes" and prefer something more along the lines of "model filling of supraglacial lakes".
- Page 9, line 7 – I dislike the term 'confirming', and would prefer something to the effect of 'consistent with findings indicating that catchment size controls surface meltwater routing delays'
- Figure S1 is redundant with Figures 2 and 3
- Figure S4, column 2 – the negative in the $m^3 s^{-1}$ is not showing up.

---

## Editor Decision (ED1)

**Editor report for tc-2019-255: Inter-comparison of surface meltwater routing models for the Greenland Ice Sheet and influence on subglacial effective pressures; Kang Yang, Aleah Sommers, Lauren C. Andrews, Laurence C. Smith, Xin Lu, Xavier Fettweis, and Manchun Li**

Dear authors,

Thank you for your patience in the long review process of the revised version of your article. As you will see, the initial reviewers disagreed on the best course of action for the article following your changes. The reviewer with supraglacial expertise recommended acceptance, whereas the reviewer with subglacial modelling expertise was not convinced of the validity of some of the subglacial experiments. I therefore sent the paper on to an extremely experienced subglacial modeller for a third opinion. They also share the concerns of the second reviewer, but agree with myself and the first reviewer on the elegance and importance of the supraglacial work. They suggest a number of different options, as you will note in their comments.

I would like to invite you to resubmit the paper with minor corrections, which are suggested by all three reviewers. Referee #2 gives some minor technical corrections which are easily addressed. Reviewer #1 suggests some amendments to the Figures. Reviewer #1 also has some significant concerns regarding the setup of the subglacial modelling. Reviewer #3 has considered this section in detail, and offers the following three solutions in their review:

1.  Omit the subglacial drainage modelling from the paper. I can imagine most or all of the reasons the authors might argue strongly against this.
2.  Perform additional model tests (see text in the review) to assess the role of parameter choices and boundary conditions in determining the model results and revise accordingly. One would like to see that the similarities/differences between the tests with different surface water routing schemes is robust to uncertain model parameter values and treatment of boundary conditions on the subglacial drainage model. This may involve replacing the current results in the paper with updated results.
3.  Change none of the model results but revise the text to dial-down results/conclusions about subglacial hydrology. This option would still benefit from additional model tests being done (even if not shown), to increase the readers' confidence in the robustness of the results/conclusions. As I imagine this is the most appealing option, I will be specific (examples given in the review).

I invite you to consider these options. If you chose Option 3, I would like to recommend paying close attention to the wording of the subglacial modelling results, especially when exploring the impact of boundary conditions on the modelled results. The changes you made following the previous review have certainly improved the validity, but both subglacial modelling experts have minimal confidence in the results as they are now presented. Reviewer #3 offers constructive options that will also address some of the concerns of Reviewer #1. Please consider the comments of all reviewers in your response.

Yours sincerely,

Dr Liz Bagshaw

Editor, The Cryosphere

---

## Editor Decision (ED2)

**Editor report Yang et al. tc-2019-255**

Dear Authors,

Thank you for your considered response to the detailed reviews. I find the paper much improved, and believe that you have sufficiently addressed the concerns of the subglacial modellers by slightly changing the focus and toning down the interpretation of the subglacial modelling results.

For example, I think this paragraph effectively informs the reader on the validity of the modelling results, and demonstrates that it should be taken as a conceptual experiment rather than a representation of real-world conditions: 'As such, our results should not necessarily be extrapolated to infer large-scale or seasonal evolution of the subglacial hydrologic system in response to different surface forcings; however, the results do provide insight into the potential diurnal sensitivity of the subglacial system to changes in supraglacial meltwater routing and the associated modification of the discharge hydrograph. For example, the amplitude of diurnal effective pressure variation for a particular day may range from < 0.5 MPa to > 3.0 MPa in this experimental setup, depending on the surface method used (Figure 3). We hope this will serve as inspiration for future work to pursue broader-scope simulations to thoroughly investigate the larger- and longer-scale effects of surface input variations on effective pressure and ice dynamics.'

However, your results do raise some interesting questions about modelling practices that deserve wider attention. This statement is the crux of the paper for me: 'Many subglacial hydrology models commonly invoke a numerical term (the "englacial void ratio") to represent englacial storage in order to provide short term storage and release of meltwater that cannot be accommodated rapidly within the subglacial system, in the absence of more realistic representation of supraglacial and englacial storage (Hewitt, 2013; 10 Werder et al., 2013; Hoffman et al., 2016). In SHAKTI, this englacial void ratio is also included as an option (Sommers et al., 2018), but our simulations considered here do not employ this term in the equations. Our results present that the supraglacial hydrologic system acts as short-term storage for surface-derived meltwater, as exhibited by the time lag of moulin inputs between models; therefore, application of an appropriate surface meltwater routing scheme may reduce the dependence of some subglacial models on a somewhat arbitrary englacial storage term to produce realistic diurnal effective pressure variations and timing lags (Werder et al., 2013; Hoffman et al., 2016).'

I therefore believe that the paper should be published, since this is an important result likely of interest to many in the subglacial modelling community as well as the wider glacier hydrology readership.

Before publication, I have a few small final amendments to request:

*Abstract:* I find this a little over-long now and wonder if you could make it slightly more concise. I also wonder if you would care to include 'Our results present that the supraglacial hydrologic system acts as short-term storage for surface-derived meltwater' as a conclusion to the subglacial modelling section? This for me is the most exciting part of your paper!

*P2, L18:* In common with Reviewer 1, I also dislike 'very long-term' since it is subjective. Instead, just use 'longer-term'.

*P8, L29:* 'As with all subglacial hydrology modelling efforts, some of these parameters are highly uncertain and poorly constrained'

I dislike this statement. I would suggest rewording to be specific to *your* model setup, rather than casting aspersions at all subglacial models. For example: 'Certain parameters are highly uncertain (for example, bed bump height and bed bump spacing), so we select values used in earlier work with SHAKTI (Sommers et al., 2018) and in the Subglacial Hydrology Model Inter-comparison Project (de Fleurian et al., 2018).'

*P16, L23:* 'As with other models, the specific effective pressures presented here are admittedly influenced by uncertain modelling parameters and boundary conditions, and should be viewed as a comparison between methods, not as a representation of real behaviour at the bed in these catchments.'

I suggest removing 'as with other models', but otherwise this statement is very useful.

*P17, L6:* 'significantly small peak' – do you mean 'smaller'?

Thank you for your contribution to the Cryosphere, and your full engagement with the review process.

Best regards,

Elizabeth Bagshaw

---

## Author Response (AR3)

August 20, 2020

Dear Editor Elizabeth Bagshaw,

Thank you for your letter on August 18 inviting revisions to our manuscript, *"Inter-comparison of surface meltwater routing models for the Greenland Ice Sheet and influence on subglacial effective pressures."* We are pleased to state that we have complied with all of the requests made by you. A stepwise, detailed response to all comments is as follows:

**Handling Editor:**

Thank you for your considered response to the detailed reviews. I find the paper much improved, and believe that you have sufficiently addressed the concerns of the subglacial modellers by slightly changing the focus and toning down the interpretation of the subglacial modelling results.

For example, I think this paragraph effectively informs the reader on the validity of the modelling results, and demonstrates that it should be taken as a conceptual experiment rather than a representation of real-world conditions: 'As such, our results should not necessarily be extrapolated to infer large-scale or seasonal evolution of the subglacial hydrologic system in response to different surface forcings; however, the results do provide insight into the potential diurnal sensitivity of the subglacial system to changes in supraglacial meltwater routing and the associated modification of the discharge hydrograph. For example, the amplitude of diurnal effective pressure variation for a particular day may range from < 0.5 MPa to > 3.0 MPa in this experimental setup, depending on the surface method used (Figure 3). We hope this will serve as inspiration for future work to pursue broader-scope simulations to thoroughly investigate the larger- and longer-scale effects of surface input variations on effective pressure and ice dynamics.'

However, your results do raise some interesting questions about modelling practices that deserve wider attention. This statement is the crux of the paper for me: 'Many subglacial hydrology models commonly invoke a numerical term (the "englacial void ratio") to represent englacial storage in order to provide short term storage and release of meltwater that cannot be accommodated rapidly within the subglacial system, in the absence of more realistic representation of supraglacial and englacial storage (Hewitt, 2013; 10 Werder et al., 2013; Hoffman et al., 2016). In SHAKTI, this englacial void ratio is also included as an option (Sommers et al., 2018), but our simulations considered here do not employ this term in the equations. Our results present that the supraglacial hydrologic system acts as short-term storage for surface-derived meltwater, as exhibited by the time lag of moulin inputs between models; therefore, application of an appropriate surface meltwater routing scheme may reduce the dependence of some subglacial models on a somewhat arbitrary englacial storage term to produce realistic diurnal effective pressure variations and timing lags (Werder et al., 2013; Hoffman et al., 2016).'

I therefore believe that the paper should be published, since this is an important result likely of interest to many in the subglacial modelling community as well as the wider glacier hydrology

readership.

**Reply:** Thank you very much for your comment. On behalf of all authors, we really appreciate all your and the three reviewers' help with the manuscript revision.

Before publication, I have a few small final amendments to request:

Abstract: I find this a little over-long now and wonder if you could make it slightly more concise. I also wonder if you would care to include 'Our results present that the supraglacial hydrologic system acts as short-term storage for surface-derived meltwater' as a conclusion to the subglacial modelling section? This for me is the most exciting part of your paper!

**Reply:** We have revised the abstract carefully to make it more concise, as suggested (p. 1, lines 14-33; p. 2, lines 1-9). "The supraglacial hydrologic system acts as short-term storage for surface-derived meltwater" is an insightful and important conclusion so we have added it to the abstract, as suggested (p. 2, line 8).

P2, L18: In common with Reviewer 1, I also dislike 'very long-term' since it is subjective. Instead, just use 'longer-term'.

**Reply:** Changed as requested (p. 2, line 24).

P8, L29: 'As with all subglacial hydrology modelling efforts, some of these parameters are highly uncertain and poorly constrained'

I dislike this statement. I would suggest rewording to be specific to your model setup, rather than casting aspersions at all subglacial models. For example: 'Certain parameters are highly uncertain (for example, bed bump height and bed bump spacing), so we select values used in earlier work with SHAKTI (Sommers et al., 2018) and in the Subglacial Hydrology Model Inter-comparison Project (de Fleurian et al., 2018).'

**Reply:** Thank you very much. We have reworded this sentence as you suggested (p. 8, lines 32-33; p. 9, lines 1-2).

P16, L23: 'As with other models, the specific effective pressures presented here are admittedly influenced by uncertain modelling parameters and boundary conditions, and should be viewed as a comparison between methods, not as a representation of real behaviour at the bed in these catchments.'

I suggest removing 'as with other models', but otherwise this statement is very useful.

**Reply:** Changed as requested (p. 16, line 26).

P17, L6: 'significantly small peak' – do you mean 'smaller'?

**Reply:** Yes, we mean "smaller". This word has been changed as suggested (p. 17, line 8).

Thank you for considering this manuscript for publication in **_The Cryosphere_**. If we may provide any additional information about the dataset or analysis, please do not hesitate to contact us at yangkangnju@gmail.com.

Respectfully submitted,

Kang Yang

Associate Professor

[revised manuscript text omitted]